# Rapid host strain improvement by in vivo rearrangement of a synthetic yeast chromosome

B. A. Blount [1,2], G-O. F. Gowers[1,2], J. C. H. Ho [1,3], R. Ledesma-Amaro[1,2], D. Jovicevic[1,2], R. M. McKiernan[1,3], Z. X. Xie[4,5], B. Z. Li[4,5], Y. J. Yuan [4,5] & T. Ellis [1,2]

Synthetic biology tools, such as modular parts and combinatorial DNA assembly, are routinely used to optimise the productivity of heterologous metabolic pathways for biosynthesis or substrate utilisation, yet it is well established that host strain background is just as important for determining productivity. Here we report that in vivo combinatorial genomic rearrangement of *Saccharomyces cerevisiae* yeast with a synthetic chromosome V can rapidly generate new, improved host strains with genetic backgrounds favourable to diverse heterologous pathways, including those for violacein and penicillin biosynthesis and for xylose utilisation. We show how the modular rearrangement of synthetic chromosomes by SCRaMbLE can be easily determined using long-read nanopore sequencing and we explore experimental conditions that optimise diversification and screening. This synthetic genome approach to metabolic engineering provides productivity improvements in a fast, simple and accessible way, making it a valuable addition to existing strain improvement techniques.

[1] Imperial College Centre for Synthetic Biology, Imperial College London, London SW7 2AZ, UK. [2] Department of Bioengineering, Imperial College London, London SW7 2AZ, UK. [3] Department of Life Sciences, Imperial College London, London SW7 2AZ, UK. [4] Key Laboratory of Systems Bioengineering (Ministry of Education), School of Chemical Engineering and Technology, Tianjin University, 300072 Tianjin, PR China. [5] SynBio Research Platform, Collaborative Innovation Center of Chemical Science and Engineering (Tianjin), Tianjin University, 300072 Tianjin, PR China. Correspondence and requests for materials should be addressed to T.E. (email: t.ellis@imperial.ac.uk)

The transplantation of pathways encoding phenotypes of interest into a more easily engineered host organism is a common process in metabolic engineering, biotechnology and synthetic biology. Optimisation of these heterologous pathways is typically achieved by determining the combination of pathway enzymes and promoter sequences that ensure their ideal expression levels[1–3]. Alternatively, pathway productivity can also be enhanced by changing the host cell within which the pathway is active, either by switching to an industrial strain or by deleting or overexpressing native genes, such as those involved in primary metabolism[4,5]. Strain improvement via changes to the host cell genome can be either rational or model-guided[6] or can use adaptive or directed evolution to promote an improved phenotype[7]. In commonly used industrial organisms like yeast, one can also make use of the many host diversification approaches currently available such as mutagenesis and large-scale sexual mating[8].

The recent arrival of *Saccharomyces cerevisiae* yeast strains where natural chromosomes are replaced by designed, synthetic chromosomes[9,10] now offers a radically new form of genome diversification that can be induced in vivo and leads to combinations of genes being deleted or altered in their expression. This is afforded by the Synthetic Chromosome Rearrangement and Modification by LoxP-mediated Evolution (SCRaMbLE) system that has been designed into the synthetic sequence of chromosomes produced by the Sc2.0 project[11]. SCRaMbLE is a combinatorial rearrangement and gene deletion system that utilises the placement of symmetrical loxP recombination sites in the 3′ untranslated regions (3′UTR) of all non-essential genes[12]. When Sc2.0 chromosomes are exposed to Cre recombinase in vivo, rearrangements occur that quickly result in major topological changes to the synthetic chromosomes within that culture[13,14].

A previous study on SCRaMbLE used genome sequencing to reveal the typical outcomes when a 100 kb circular region of synthetic chromosome DNA is rearranged in yeast grown in standard growth conditions[13]. Owing to the complexity of some recombination events (e.g., tandem duplications), the post-SCRaMbLE DNA topology could not be determined for all sequenced strains using only short-read technology. However, in those strains where sequencing did resolve the new layout, extensive rearrangements were observed, both in cells that maintained normal growth and those exhibiting impaired growth.

The diversity of outcomes seen just with this short synthetic DNA region highlights how millions of unique genotypes could be produced by SCRaMbLE of synthetic chromosomes within a population of cells. We reasoned that this would offer a huge potential phenotype space that could be exploited for strain improvement, and specifically for generating strains with altered genetic backgrounds that provide a benefit for heterologous pathways of industrial relevance. Thus we set out to develop an approach that combines the chromosomal diversification seen previously with SCRaMbLE, with the expression and screening of heterologous pathways hosted in synthetic yeast. To go beyond previous work and scale-up the potential phenotypic space, we used yeast now containing a fully synthetic 536 kb chromosome, and to ensure that the rearrangements could be easily mapped, we sequenced resulting strains of interest using long-read nanopore sequencing. Our work describes SCRaMbLE as a powerful method for host strain diversification. When haploid synthetic yeast strains expressing heterologous pathways are subject to a short SCRaMbLE process with no applied selective pressure, diverse genetic backgrounds are quickly produced that significantly improve the productivity of multiple heterologous metabolic pathways of industrial interest.

## Results

**SCRaMbLE generates a strain with improved biosynthesis yield.** To investigate whether induced SCRaMbLE of a synthetic chromosome in vivo can enhance a host for biosynthesis, we assessed production of a secondary metabolite using the pigment-producing violacein biosynthesis pathway from *Chromobacterium violaceum*[15]. An exemplar synthetic biology study on heterologous expression optimisation was recently able to achieve a two-fold increase in yield from this pathway in *S. cerevisiae* using model-guided iterative design[16]. A 2μ plasmid, pJCH017, was constructed to express the five pathway genes (Supplementary Fig. 1). To minimise the effects of any criteria other than host genome SCRaMbLE, no loxP sites were included in the plasmid and all genes were assembled with strong promoters. pJCH017 and the Cre recombinase expression plasmid, pSCW11-creEBD[11], were transformed into synV, a haploid yeast strain in which the natural chromosome V has been replaced with a synthetic version containing 174 loxPsym sites[17]. SCRaMbLE was then induced in a growing culture of this strain for 4 h and cells plated onto SDO URA⁻ agar medium to maintain selection for the violacein plasmid. Selection for the Cre recombinase plasmid was not maintained post-SCRaMbLE.

To screen for violacein production, 87 of the resultant colonies, along with un-SCRaMbLEd controls, were picked and grown to an optical density between 0.5 and 0.7 before spotting 10 μl of each culture onto SDO URA⁻ agar medium (Fig. 1a, b). Of these spotted cultures, the strain in position B2 (named VB2) was visibly darker than controls and thus likely to be producing more of the violacein pigment molecule. Violacein production by VB2 was compared to that of the synV-pJCH017 strain by colourimetric assay and was found to be 2.3 times higher (Fig. 1c). VB2 cells cured of the plasmid and retransformed with pJCH017 from the original DNA stock retained the high production, ruling out plasmid-based mutations as an explanation for the enhanced phenotype (VB2c-pJCH017, Fig. 1c). Microscopic analysis of VB2 cells revealed these to be typically larger than synV equivalents, with this increased size more pronounced in rich medium than in synthetic medium (Supplementary Figs. 2, 3).

To determine whether the enhanced violacein production is related to the pathway metabolism or to increased expression of the pathway enzymes, cured synV and VB2 stains were next transformed with a 2μ vector expressing superfolder green fluorescent protein (sfGFP) and a *URA3* auxotrophic marker (pBAB012). VB2-pBAB012 showed enhanced sfGFP production, compared to synV-pBAB012, supporting the case for VB2 providing a general boost in the expression of genes from the 2μ vectors (Fig. 1d, Supplementary Fig. 4–6). To determine whether this effect is 2μ replicon specific, fluorescence from sfGFP expressed from a *CEN6*/ARS replicon (pBAB011) was also assessed in both VB2 and synV hosts (Fig. 1e) revealing no increase in expression. To further rule out *URA3*-specific factors, sfGFP expression was measured with *LEU2* auxotrophic marker vectors giving a similar pattern of results (Fig. 1e). Lower fluorescence output was seen from plasmids with a *LEU2* marker, consistent with previous findings[18].

Quantitative PCR (qPCR) measurement of the *kanR*, *vioB* and *vioD* genes on pJCH017 next determined the relative copy numbers of violacein pathway plasmid in VB2-pJCH017 and synV-pJCH017 (Fig. 1f). This revealed a 1.89-fold increase in 2μ DNA copies per genomic DNA in VB2 cells compared to synV, and this increased plasmid copy number likely explains the observed increase in violacein production in this strain.

To demonstrate that the increase in plasmid copy number in VB2 can also enhance biosynthesis from other 2μ constructs, a pathway for penicillin G production in *S. cerevisiae* was next assessed. The best-performing pathway constructs from a

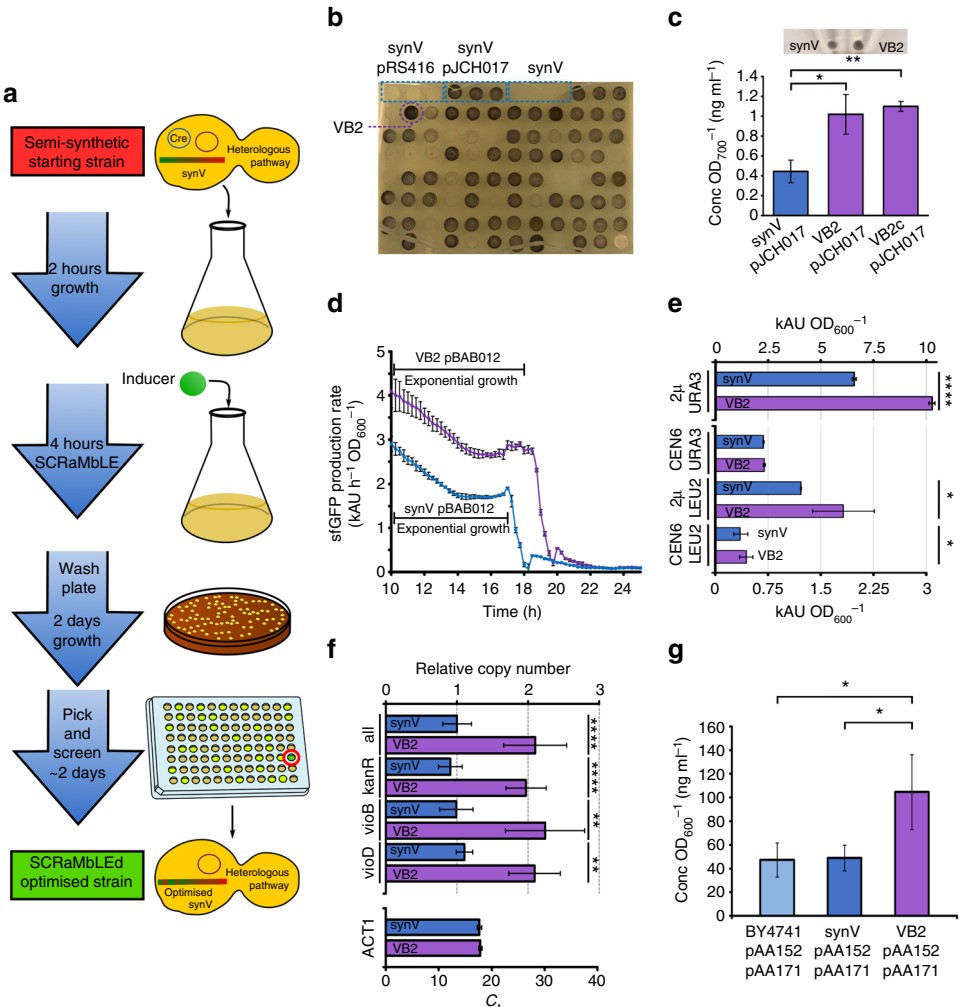

**Fig. 1** Optimising a synV host by SCRaMbLE for enhanced violacein production. **a** The workflow of a SCRaMbLE host optimisation process. **b** Eighty seven post-SCRaMbLE synV-pJCH017 colonies and synV, synV-pRS416 and synV-pJCH017 controls were grown in liquid culture, spotted onto SDO URA⁻ agar medium and grown for 2 days. The best performing strain, VB2, is highlighted. **c** Results of colourimetric determination of violacein yields from synV-pJCH017, VB2-pJCH017 and VB2c-pJCH017 cultures, $n = 3$. Inset are microtubes in which equal volumes of synV-pJCH017 and VB2-pJCH017 culture, normalised for $OD_{700}$, have been pelleted by centrifugation. **d** sfGFP production rates from synV-pBAB012 and VB2-pBAB012 cultures, $n \geq 7$. Data shown are the increase in 520 nm fluorescence value over the previous hour normalised to the $OD_{600}$ value at the midpoint of that hour. **e** The 520 nm fluorescence levels of cultures of synV and VB2 strains containing pBAB011 (CEN6/URA3), pBAB012 (2μ/URA3), pBAB015 (CEN6/LEU2) or pBAB016 (2μ/LEU2) plasmids after 29.5 h of growth. Fluorescence values are normalised for $OD_{600}$, $n \geq 7$. **f** qPCR evaluation of pJCH017 copy number in total DNA preparations from stationary synV-pJCH017 and VB2-pJCH017 cultures. The vioD, vioB and kanR targets are located on pJCH017 and 'all' represents the combined data from amplification of these three loci. Relative copy number is the calculated concentration of pJCH017 in the total DNA for each experiment normalised to the combined synV-pJCH017 value. The ACT1 control shows the threshold cycle for amplification of a chromosomal ACT1 target for each DNA preparation, demonstrating equal levels of genomic DNA, $n = 6$ (2 technical replicates each of 3 biological replicates). **g** The amount of penicillin G secreted by BY4741-pAA152/171, synV-pAA152/171 and VB2-pAA152/171 strains as determined by LC-MS. Values are normalised for $OD_{600}$, $n = 3$. All values plotted are mean averages and error bars represent 1 standard deviation from the mean. Replicate numbers represent biological replicates except where otherwise stated. Asterisks denote two-tail p-value as determined by two-sample t-test, with *$p \leq 0.05$, **$p \leq 0.01$, and ****$p \leq 0.0001$

previous study, encoded on a 2μ and a CEN6 vector, were co-transformed into both synV and VB2 hosts[19]. These strains, along with an existing BY4741 strain containing the pathway plasmids, were then assayed for secreted penicillin yields by liquid chromatography–mass spectrometry (LC-MS). Penicillin production was indeed enhanced by 2.1-fold in the VB2 strain compared to that produced by synV or BY4741 hosts (Fig. 1g, Supplementary Fig. 7). The penicillin G pathway has proved difficult to express in yeast, requiring subcellular localisation of pathway enzymes and library-based screening strategies to achieve bioactive levels of the antibiotic molecule[19]. The yield achieved with the VB2 strain, 14.9 ng ml⁻¹, is significantly higher than the

previous highest recorded yields between 5 and 6 ng ml⁻¹ [19]. Thus VB2, generated by a short SCRaMbLE protocol with no selective pressure, now provides a new S. cerevisiae host capable of increasing the expression of pathways and genes on 2μ plasmids.

**SCRaMbLE to generate a strain with enhanced growth on xylose.** In addition to engineering S. cerevisiae with heterologous genes to produce biomolecules of interest or value, altering cellular metabolism to allow or enhance growth on alternative energy sources is often desirable. To demonstrate that SCRaMbLE

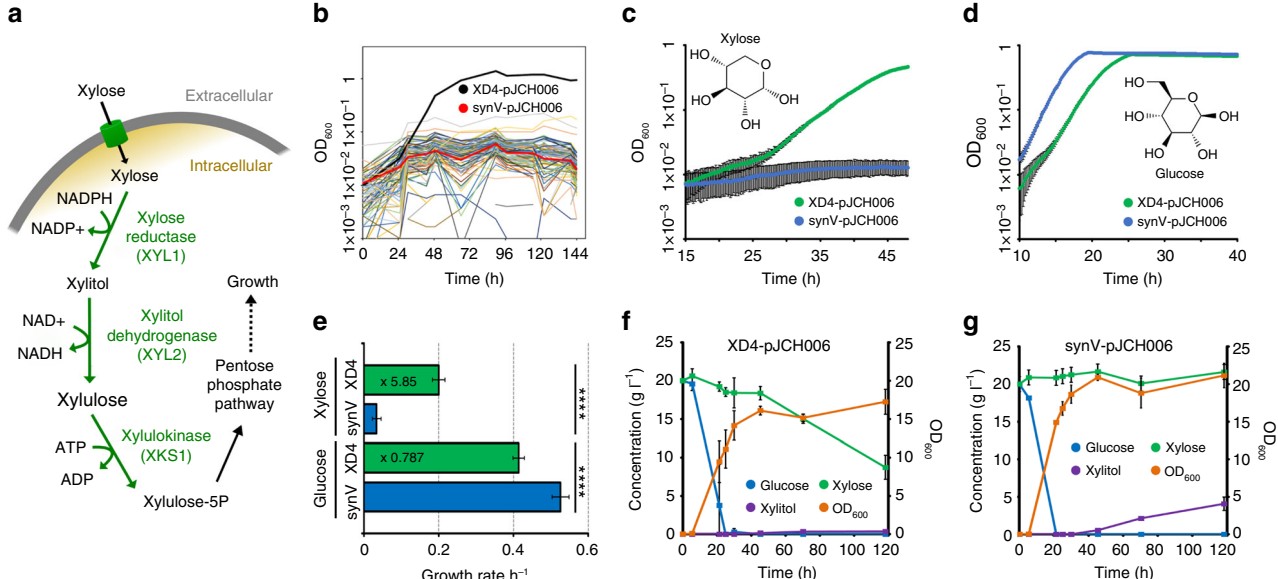

**Fig. 2** SCRaMbLE generates a host strain specialised for xylose utilisation. **a** Xylose oxidoreductase pathway through which xylose is converted to xylulose-5 phosphate before entering the pentose phosphate pathway. Enzymes with names in green are encoded by genes on pJCH006. **b** $OD_{600}$ values for the initial 96-well based screening of colonies in SCX URA$^-$ medium. The synV-pJCH006 control and the colony in well D4 are highlighted, $n = 1$. **c** The growth of XD4-pJCH006 and synV-pJCH006 in synthetic xylose medium over 48 h in 96-well format, $n = 19$. **d** The growth of XD4-pJCH006 and synV-pJCH006 in synthetic glucose medium over 48 h in 96-well format, $n \geq 18$. **e** The growth rate per hour of XD4-pJCH006 and synV-pJCH006 in synthetic xylose and synthetic glucose media in 96-well format. Rates were calculated over 5 h of exponential growth, $n \geq 18$. **f** HPLC-derived concentrations of glucose, xylose and xylitol and the optical density ($OD_{600}$) of 25 ml XD4-pJCH006 cultures in SC URA$^-$ medium with 2% glucose and 2% xylose over time, $n = 2$. **g** The equivalent data derived from synV-pJCH006 cultures, $n = 2$. All values plotted are mean averages and error bars represent 1 standard deviation from the mean. Replicate numbers represent biological replicates. Asterisks denote two-tail $p$-value as determined by two-sample $t$-test, with ****$p \leq 0.0001$

can also improve the growth of yeast in an attractive alternative carbon source, the heterologous pathway of xylose utilisation was introduced (Fig. 2a). Xylose is a much-targeted alternative carbon source for yeast growth, due to its high abundance in lignocellulosic biomass[20]. As *S. cerevisiae* is unable to grow using xylose as the sole carbon source, plasmid pJCH006 containing the *XYL1* and *XYL2* genes from *Scheffersomyces stipitis* and an additional copy of the *XKS1* xylulokinase from *S. cerevisiae* (Supplementary Fig. 8), was constructed and transformed into the synV strain to provide the oxidoreductase pathway for xylose utilisation[21].

SCRaMbLE was induced, as before, for 4 h with synV-pJCH006/pSCW11-*creEBD* cells and these were plated onto SDO URA$^-$ agar medium. Eighty seven colonies were picked at random and grown at 30 °C in a microwell plate in selective media with xylose as the sole carbon source (SCX URA$^-$). Optical densities of cultures were monitored over 5 days to identify colonies showing improved growth in xylose compared to synV-pJCH006 controls. Strain XD4 was selected from the SCRaMbLE colonies as it exhibited dramatically enhanced growth in xylose in this preliminary screen (Fig. 2b). To further characterise XD4, plate-based growth assays were performed comparing XD4-pJCH006 to synV-pJCH006 in xylose- and glucose-containing media (Fig. 2c, d). XD4-pJCH006 shows on average 5.85 times the growth rate of synV-pJCH006 in SCX URA$^-$ xylose medium, whilst having only 0.79 times the growth rate in SDO URA$^-$ glucose medium (Fig. 2e). A similar increase in growth rate in xylose medium was also observed when the strains were grown in 60 ml cultures in baffled flasks (Supplementary Fig. 9). Of the 19 XD4-pJCH006 cultures assayed in 96-well plate format, the average growth rate was $0.2\,h^{-1}$, with some cultures having a growth rate over 5 h measuring as high as $0.238\,h^{-1}$. This compares favourably to the $0.18\,h^{-1}$ maximum growth rate

reported in similar conditions after extensive optimisation of this pathway in the diploid CEN.PK 113-5D strain and the $0.21\,h^{-1}$ recently achieved in CEN.PK 113-7D via targeted overexpression and mutation strategies[22,23].

Mutations in pJCH006 were discounted as being responsible for the phenotype by curing XD4 of this plasmid and retransforming it with original pJCH006 DNA stock. After 48 h growth in xylose media in 96-well format, the endpoint optical density of this retransformed strain (XD4c-pJCH006) was equivalent to that of XD4-pJCH006 (Supplementary Fig. 10). Microscopy of XD4 cells also revealed no noticeable morphological differences to synV in SX URA$^-$ medium (Supplementary Figs. 11, 12).

To further investigate the XD4 phenotype, 25 ml cultures of XD4-pJCH006 and synV-pJCH006 with 4% xylose, 4% glucose or 2% xylose plus 2% glucose were grown in flasks for 5 days with samples periodically taken and analysed by high-performance liquid chromatography (HPLC) (Fig. 2f, g, Supplementary Fig. 13). These analyses show that, in the 4% xylose cultures, XD4-pJCH006 removes around 13.5 g of xylose from the medium, whereas synV-pJCH006 does not remove a detectable amount. In the 2% xylose plus 2% glucose medium, the synV-pJCH006 strain accumulates xylitol, whereas this is not the case with XD4-pJCH006. The oxidoreductase pathway used here is known to generate a redox imbalance in *S. cerevisiae* that results in xylitol accumulation and it seems that XD4 is able to bypass this[24].

**Long-read sequencing to determine SCRaMbLE rearrangements.** In order to confirm that SCRaMbLE events have occurred and determine what type of chromosome rearrangements have generated the phenotypes of interest, it is desirable to sequence the genomes of chosen strains. While short-read technologies are

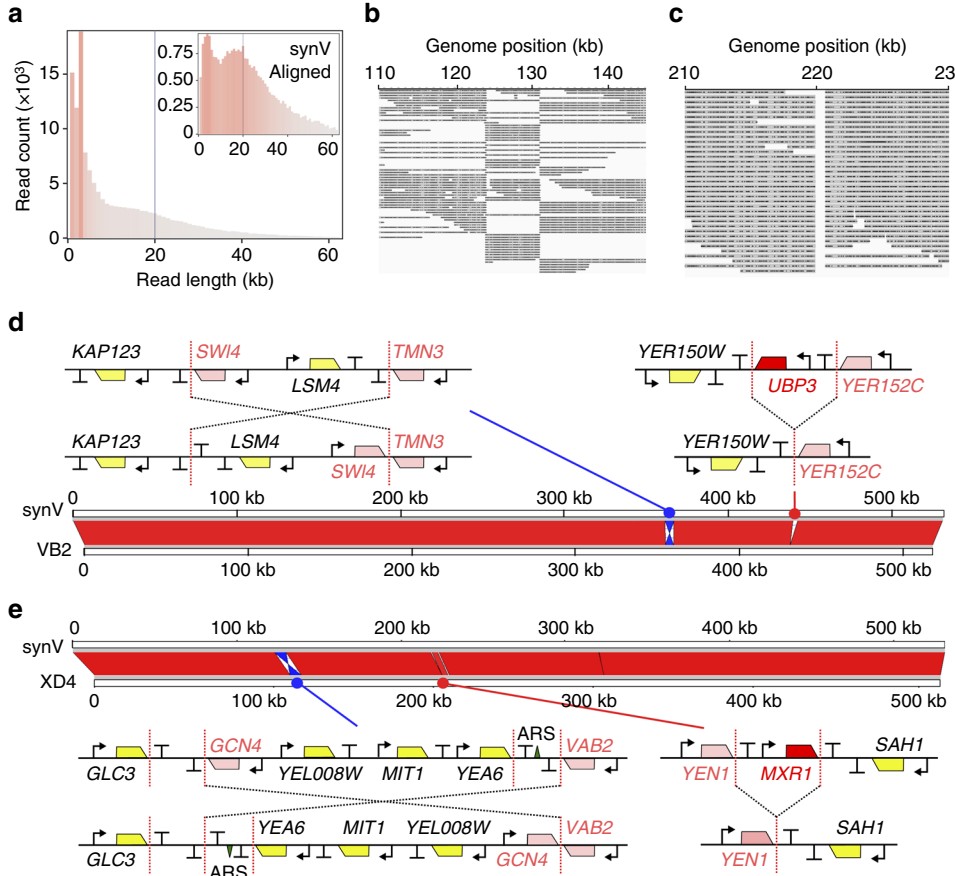

**Fig. 3** Nanopore sequencing to identify the SCRaMbLE events in enhanced strains. **a** Base-called read length distribution of the XD4 nanopore sequencing run. The line at 20 kbp indicates the DNA shear length during library preparation. Inset is the same analysis of the subset of the data that aligned by LAST to the synthetic chromosome V sequence. **b** An alignment of XD4 nanopore sequencing reads to parental synV sequence around the identified inversion region. **c** An alignment of XD4 nanopore sequencing reads to parental synV sequence around the identified MXR1 deletion region. **d** Full alignment of the VB2 contig, and **e** the alignment of the XD4 contig to the parental synV. The inversion and deletion regions are expanded to show diagrammatic interpretations of the SCRaMbLE events, showing pre- (top) and post-SCRaMbLE (bottom) configurations, with locations of promoters (arrows), coding sequences (polygons), terminators (T), autonomous replicating sequences (ARS, triangle) and loxPsym sites (red dotted lines) indicated. Red coding sequences are deleted by SCRaMbLE events and pink coding sequences have their 3′ UTR regions altered

now standard for genome sequencing, these are optimised to identify nucleotide changes and not the kind of chromosomal rearrangements generated by SCRaMbLE. Indeed, in a previous study sequencing post-SCRaMbLE rearrangements of a short circular synthetic yeast chromosome arm, Illumina-based sequencing was not able to resolve all sampled strains[13]. Recent improvements in rapid long-read sequencing technologies, like nanopore sequencing, now offer an alternative that addresses this.

To demonstrate the use of nanopore sequencing to determine post-SCRaMbLE chromosomes, genomic DNA preparations from XD4 and VB2 were sequenced, each on a single R9 flow cell using the portable ONT MinION device. For VB2, the single 48 h run yielded 2.83 Gbp of base-called sequence data with a mean average read length of 11 kb and an n50 of 16.3 kb. This resulted in a ~233× average genome coverage when reads below 1 kb were excluded. For XD4, the single 48 h run yielded 1.72 Gbp of base-called sequence data with a read-length distribution with a mean average of 12.5 kb and an n50 of 24 kb, corresponding to an average genome coverage of ~133× when reads below 1 kb were excluded (Fig. 3a). Reads were corrected using Canu v1.5[25] and filtered to yield reads between 1 kb and 60 kb in length (Fig. 3b, c). Smartdenovo was then used to assemble this set of reads into de novo contiguous sequences (contigs). Single contigs were aligned against the parental synV sequence and visualised using

the Integrative Genome Viewer[26]. For both strains, no significant rearrangements or copy number variations were seen for all non-synthetic chromosomes.

The sequence of VB2 shows that two SCRaMbLE events have occurred in synthetic chromosome V, deleting the 2971 bp region encoding UBP3 and inverting the 5337 bp region containing the SWI4 and LSM4 coding sequences (Fig. 3d). In this inversion event, LSM4 retains its 3′UTR but SWI4 does not. The gene TMN3 also loses its 3′UTR. The recombination sites all map to loxPsym sites, confirming that SCRaMbLE was responsible for the chromosomal changes. These SCRaMbLE events were further confirmed by PCR screening (Supplementary Fig. 14). Sequencing of XD4 also revealed just two SCRaMbLE events in synthetic chromosome V, an inversion of a 7 kb locus encoding GCN4, YEL008W, MIT1 and YEA6 genes (plus ARS510), and deletion of a 785 bp region containing the short MXR1 coding sequence (Fig. 3b, c, e). As with the VB2 sequencing, these recombination events mapped to loxPsym sites and were confirmed by PCR amplification with region-specific primers (Supplementary Fig. 15).

Deletions of UBP3 and MXR1 were recreated in BY4741 using CRISPR-mediated recombination, to determine whether these events lead to the VB2 and XD4 phenotypes, respectively. The BY4741 ubp3Δ-pBAB012 strain showed no increase in sfGFP

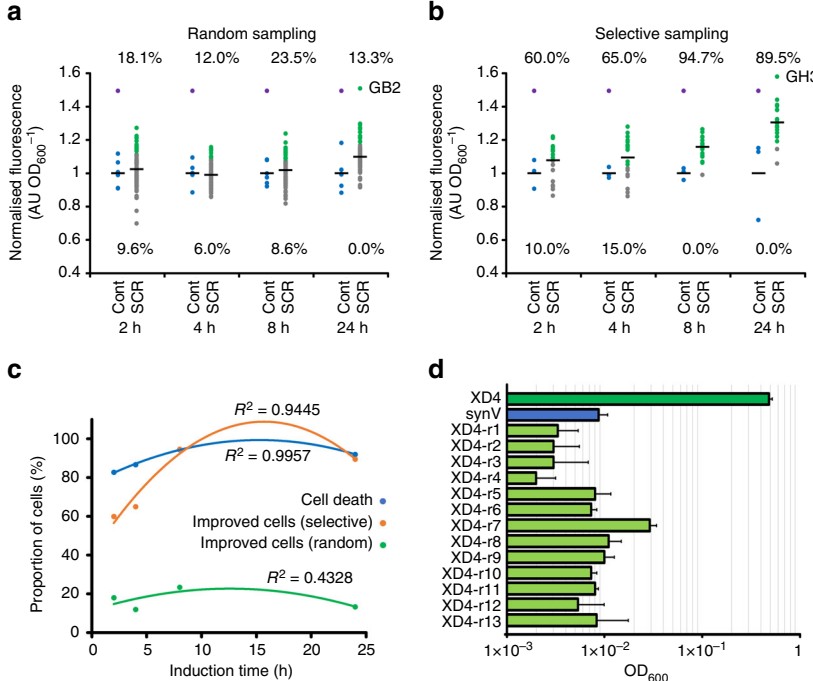

**Fig. 4** Investigating the dynamics of SCRaMbLE. **a** The endpoint 520 nm fluorescence values of SCRaMbLEd synV-pBAB016 colonies picked randomly after 2, 4, 8 or 24 h of β-estradiol induction and then grown in 96-well plate format for 24 h. For each induction length, ≥81 randomly selected induced colonies were characterised and 6 colonies were also characterised from an uninduced synV-pBAB016 control plate. Fluorescence values were normalised for culture $OD_{600}$ and are given as a proportion of the mean un-SCRaMbLEd control fluorescence value for that induction length. Blue dots denote un-SCRaMbLEd controls (cont), grey dots denote SCRaMbLEd cells (SCR) that have a fluorescence value equal to or lower than the highest un-SCRaMbLEd control, green dots denote SCRaMbLEd cells that have a higher fluorescence values than any of the un-SCRaMbLEd controls and purple dots denote the relative VB2-pBAB016 fluorescence value from Fig. 1e. Percentage values above the plotted data are the proportion of SCRaMbLEd cells that have a higher fluorescence value than any of the un-SCRaMbLEd controls and values below the plotted data give the proportion of SCRaMbLEd cells with a lower fluorescence value than those controls. Black horizontal lines denote the mean average relative fluorescence of that group of cells, excluding the VB2-pBAB016 value in the case of the controls. **b** The equivalent information but with SCRaMbLEd colonies visually screened under blue light to selectively pick colonies with high fluorescence. For each induction length, ≥19 selected colonies were characterised along with 3 un-SCRaMbLEd controls. **c** For each induction length, the proportion of cell death and the proportion of improved cultures, i.e., those with higher fluorescence output than any of the equivalent un-SCraMbLEd controls, with either selective or random picking of colonies. Lines are second-degree polynomial curves of best fit, with $R^2$ values stated. **d** $OD_{600}$ achieved after 48 h growth in synthetic xylose medium of XD4-pJCH006, synV-pJCH006 and the 13 fastest growing re-SCRaMbLEd XD4-pJCH006 colonies. Values plotted are mean averages of 3 biological replicates and error bars represent 1 standard deviation from the mean

output compared to BY4741-pBAB012 (Supplementary Fig. 16), and while BY4741 *mxr1Δ*-pJCH006 did show a significant increase in xylose growth compared to BY4741-pJCH006, the improvement was not on the scale seen in XD4 (Supplementary Fig. 10). This shows that in both cases the deletions alone are not sufficient to explain the phenotypes, indicating that the more complex inversion events are contributing factors. Knowledge of the new layout of SCRaMbLEd synthetic chromosomes provides the foundations for further studies that can determine how specific rearrangements and deletions can lead to observed improvements for phenotypes of interest.

**Comparing conditions for SCRaMbLE induction and screening**. Both VB2 and XD4 strains were identified from 4 h SCRaMbLE reactions followed by screening of randomly picked colonies. We avoided applying selective pressure during SCRaMbLE and subsequent plating as selective pressures are not always available for a desired phenotype, and <100 colonies were screened each time, simply for the convenience of using 96-well plate format. The fact that improved strains were identified by this method for two quite different cases led us to further investigate the likelihood of seeing an improved strain by SCRaMbLE. For ease of analysis, we chose to screen for increased

green fluorescence when sfGFP is expressed from a 2μ plasmid, as seen previously with VB2, and we induced synV SCRaMbLE for varying times, and also screened both with and without a prior selection. β-estradiol concentration at induction was kept constant at 1 μM, as varying the inducer concentration has a limited effect on post-SCRaMbLE cell viability levels (Supplementary Fig. 17).

pSCW11-*creEBD* and pBAB016 (2μ plasmid with *sfGFP*) were transformed into synV. In quadruplicate, cultures of this strain underwent SCRaMbLE with a β-estradiol induction time of 2, 4, 8 or 24 h. Control cultures also underwent the process but without addition of the inducer. To represent no selective pressure, 83 SCRaMbLE-induced colonies and 6 uninduced colonies were randomly selected for each condition and sfGFP fluorescence per cell was quantified after 24 h (Fig. 4a). To represent selective pressure, this process was repeated but this time only taking colonies post-SCRaMbLE that, by eye, appeared to fluoresce strongly under blue light (Fig. 4b).

Colony counts from the plated SCRaMbLE cultures indicate that viability reduces with longer induction times, ranging from 17.3% with 2 h induction to 5.5% with 8 h induction (Fig. 4c), presumably as increased SCRaMbLE events lead to the deletion or inactivation of more essential genes. This trend does not continue to the 24 h induction culture, which has an 8% viability,

presumably due to a combination of the inducer degrading over time within the culture and previous viability losses being mitigated by cell growth from survivors.

With 96-well throughput, cells with similar sfGFP output to that seen with VB2 were identified by both random and selective screening (Fig. 4a, b). Selective screening markedly increased the proportion with an increased output, with 94.7% of selected 8 h induction colonies achieving this compared to only 23.5% of the randomly selected equivalents. With either approach, an 8 h induction of SCRaMbLE resulted in the highest proportion of cells with mean fluorescence above all un-SCRaMbLEd controls. As with the cell viability data, the percentage of improved strains arising rose as SCRaMbLE induction increased to 8 h but was reduced by a 24 h SCRaMbLE (Fig. 4c).

Analysis by PCR of the genomic DNA from the highest-performing sfGFP strains from the random and selective screens (GB2 and GH3, respectively) showed that neither had replicated either of the two SCRaMbLE events identified by sequence analysis of VB2, and so each had achieved increased fluorescence via different chromosome rearrangements (Supplementary Fig. 14). It is not unexpected that each SCRaMbLE will generate a unique outcome, because the total number of possible rearrangement combinations with hundreds of loxPsym sites is astronomical.

Finally, to investigate whether additional SCRaMbLE of a selected strain can further improve a phenotype, a second round of SCRaMbLE was performed on the xylose-utilising XD4 strain. XD4-pJCH006/pSCW11-creEBD cells were SCRaMbLEd with a 4 h induction, but this time plated directly onto SCX URA⁻ agar to accelerate selection. Visible colonies were not evident for 7 days and viable colony numbers were far lower than previously seen with just a primary SCRaMbLE ($7 \times 10^3$ cfu ml$^{-1}$ vs. $2.6 \times 10^5$ cfu ml$^{-1}$, Supplementary Fig. 18). The 13 largest colonies were assayed for growth in SCX URA⁻ medium but all displayed slower growth than XD4 (Fig. 4d). This indicates that the initial SCRaMbLE provided strain improvement, but further SCRaMbLE of an already SCRaMbLEd strain proved detrimental. This may be due to the reduction in possible genetic and phenotypic space that results from the isolation of a particular SCRaMbLEd strain, with further events reducing fitness, causing inviability or simply not affecting the desired phenotype. These results indicate that a longer SCRaMbLE induction would typically be a more effective strategy for strain improvement, rather than rounds of SCRaMbLE punctuated by screening and isolation of individual strains.

## Discussion

The in vivo rearrangement of synthetic yeast chromosomes offers a powerful approach for optimising the host for improved production of molecules or for enhanced growth on alternative substrates. As demonstrated here, SCRaMbLE effectively offers an extreme form of inducible mutation capable of quickly generating host strains with significantly enhanced performance for different tasks of relevance to biotechnology. The process compares favourably in several ways to existing strain improvement methods, such as transcription machinery engineering, gene deletion library screens and genome shuffling (Table 1), and can be used as an alternative to or to complement these established techniques. It is rapid, with diversification taking only a few hours, and it does not require application of a selective pressure throughout the process. Optimisation by SCRaMbLE is inexpensive and relatively non-technical, requiring minimal equipment or reagents. Additionally, when using SCRaMbLE to improve heterologous pathway performance, such as in VB2 and XD4, results are also achieved with minimal knowledge of the

pathway. Indeed, the pathway genes used in the cases described here were all paired with strong promoters with no attempts to balance their expression levels beforehand.

As with other mutation methods used to improve strains, SCRaMbLE can simply be treated as a black box process where enhanced hosts are generated without the genetic events causing the improvements being fully understood. However, as shown here, desktop long-read sequencing now makes it possible to quickly determine the layout and content of rearranged fully synthetic chromosomes. While knowledge of the genomic changes is only part of the puzzle towards explaining global outcomes such as altered metabolism, it allows us to speculate on mechanisms for the phenotypic changes and provides the foundations for future work that can fully uncover how SCRaMbLE leads to a given strain improvement.

For the two strains sequenced here, VB2 and XD4, the identified inversions and deletions point towards possible routes for increased 2μ copy number and improved xylose utilisation, respectively. For example, in the inversion event in VB2, the SWI4 gene loses its 3′UTR and thus may subsequently have altered expression or regulation. SWI4 encodes the DNA-binding component of the SBF transcription complex, which has a major role in regulating several G1-specific genes, including DNA synthesis genes[27]. As 2μ vector replication occurs in S phase, a change in the regulation of DNA synthesis genes in G1, due to changes to SWI4 expression, could be responsible for increased 2μ copy number if levels of DNA synthesis machinery are altered going into S phase[28]. For XD4, the inversion event leaves GCN4 without its standard 3′UTR sequence and so also may alter its expression or regulation. This is of particular interest given that Gcn4 is a global metabolic transcription factor and has previously been identified to be of importance for xylose utilisation[29,30]. In addition, the deletion of MXR1 may improve xylose utilisation by indirectly correcting the redox imbalance known to lead to xylitol accumulation in un-SCRaMbLEd yeast[24]. MXR1 encodes a cytosolic methionine-S-sulfoxide reductase involved in oxidative stress that is indirectly linked to NADPH via its use of the redox carrier thioredoxin[31,32].

In future work, RNAseq, proteomics and metabolomics studies of these SCRaMbLEd strains or others will help us better understand how various deletions and rearrangements on synthetic chromosomes lead to different phenotypic changes. The full effects of deletions and inversions on local gene expression may well be highly complex, as the altered topology at SCRaMbLEd loci is also likely to affect the expression of neighbouring genes through multiple mechanisms, such as changes in local nucleosome positioning, disruption of nearby promoter regulation and repositioning of genes relative to one another. Each phenotype may well be due to one or a combination of many of these various genomic changes and will be an interesting challenge to explore. It is important to also consider that these changes may have other unanticipated effects, such as reducing the fitness of the host in conditions not being screened for. As with hosts generated by other strain diversification methods, further testing of performance would be needed before using SCRaMbLE-generated strains for production at commercial scale.

The unique power of the SCRaMbLE method is in being able to generate so many different events randomly and combinatorially with ease, and offering a rapid approach for yeast host strain improvement that we show here can significantly benefit multiple heterologous pathways. Given that the synthetic yeast strains being generated by the Sc2.0 project are all freely available and have no associated intellectual property restrictions[33], host optimisation by SCRaMbLE represents a powerful and accessible strain enhancement technology with broad applications.

**Table 1 Comparison of SCRaMbLE to common *S. cerevisisae* strain improvement strategies**

| Technique | SCRaMbLE | Mass mating/ genome shuffling | Transcription machinery engineering/ rewiring | Deletion library screening | Targeted deletion/ overexpression/ underexpression | Random mutagenesis/ directed evolution |
|---|---|---|---|---|---|---|
| Strain prerequisites | Synthetic chromosomes and recombinase vectors | Natural isolates with desired properties | Efficient transformation into strain | Deletion library, efficient transformation into strain | — | — |
| Prior knowledge | — | Phenotypic analysis of strains | Identification of transcriptional regulators/ regulated promoters | — | Prior gene characterisation studies, knowledge of metabolic processes | — |
| Preparation work | Transform pathway into one strain | Transform pathway into half of mating strains | Large-scale library construction | Transform pathway into many strains | Construction of TAR cassettes | Transform pathway into one strain |
| Technical requirements | Basic strain cultivation | Strain mating/ protoplast fusion, tetrad dissection | DNA assembly, library scale DNA transformation | Library scale DNA transformation | DNA assembly/gene editing tools | Strain cultivation, chemostat/ tubidostat, mutagenesis |
| Timescale to improved strain | 1–2 weeks | Months to years | Months to years | Weeks to months | Months | Months to years |
| Strain diversity generated | Coding sequence deletions, inversions, duplications, translocations | Allele combinations, polyploidy | Novel regulation | Coding sequence deletions | Specific targeted changes | Point mutations |
| Other factors | — | High prevalence of deleterious alleles, mitotic instability | High throughput screening often necessary | Higher numbers of combined knockouts exponentially increases transformation requirements | Understanding of biological mechanisms often not sufficient for effective improvement | High number of off-target effects, selective pressure often required |

Technical requirements do not include screening methods, which are common to all methods. Timescale to improved strain estimates length of time from introduction of a heterologous pathway to isolation and characterisation of a strain with an improved phenotype

## Methods

**Strains and media**. Yeast strains BY4741 and yXZX846 (BY4741 with synthetic chromosome V, referred to as synV in the text) were generated in previous studies[17]. All other yeast strains were generated during this study. *Escherichia coli* DH10B (Life Technologies) was used as a bacterial cloning and plasmid propagation host. Luria Bertani medium was used for bacterial growth, YPD (10 g l$^{-1}$ yeast extract, 20 g l$^{-1}$ peptone, 20 g l$^{-1}$ glucose) was used for non-selective yeast growth and either synthetic glucose (SDO, 6.7 g l$^{-1}$ yeast nitrogen base, 1.4 g l$^{-1}$ yeast synthetic drop out medium supplement without histidine, leucine, tryptophan and uracil, 20 g l$^{-1}$ glucose) or synthetic xylose (SX, 6.7 g l$^{-1}$ yeast nitrogen base, 1.4 g l$^{-1}$ yeast synthetic drop out medium supplement without histidine, leucine, tryptophan and uracil, 40 g l$^{-1}$ xylose) supplemented with histidine (20 mg l$^{-1}$), leucine (120 mg l$^{-1}$), tryptophan (20 mg l$^{-1}$) and uracil (20 mg l$^{-1}$) as appropriate was used for selective growth. Yeast media components were supplied by Sigma Aldrich.

**Plasmid construction**. All plasmids generated in this study were assembled using the MoClo-Yeast Toolkit (YTK)[34]. pJCH017 was constructed by amplifying *vioA*, *vioB*, *vioC*, *vioD* and *vioE* from pJC104 and assembling into five intermediate plasmids, encoding *pTDH3-vioA-tTDH1*, *pPGK1-vioB-tENO2*, *pHHF2-vioC-tPGK1*, *pCCW12-vioD-tADH1* and *pTEF1-vioE-tSSA1*, respectively. These expression cassettes were then assembled together with a *URA3* auxotrophic marker, a 2μ yeast replicon and ColE1-kanR bacterial replicon components to yield the final plasmid. pJCH006 was constructed via 3 intermediate plasmids, encoding *pTEF2-XYL1-tADH1*, *pTEF1-XYL2-tTDH1* and *pTDH3-XKS1-tENO2* respectively. These expression cassettes were then assembled together with a *URA3* auxotrophic marker, a 2μ yeast replicon and ColE1-kanR bacterial replicon components to yield the final plasmid. sfGFP expression plasmids pBAB011 (*CEN6, URA3*), pBAB012 (2μ, *URA3*), pBAB015 (*CEN6, LEU2*) and pBAB016 (2μ, *LEU2*) were assembled combining ConLS, pTEF2, *sfGFP*, tADH1, ConR1 and ColE1-AmpR with the yeast replicon and auxotrophic marker indicated. Enzymes were supplied by New England Biolabs.

**Yeast transformations**. All yeast transformations were performed using the lithium acetate method with a 14 min 42 °C heat shock followed by a 10 min recovery in 5 mM calcium chloride prior to plating on appropriate selective media[9].

**DNA isolation**. Genomic DNA was isolated for nanopore sequencing and qPCR analysis using Genomic-tip Kits (Qiagen) using wide-bore pipette tips. Genomic

DNA was isolated for PCR analysis using the GC Prep method[35]. Plasmids were isolated from bacterial hosts using the QIAprep spin Miniprep Kits (Qiagen).

**SCRaMbLE**. Single colonies were picked and used to inoculate 5 ml of appropriate selective glucose medium and cultured overnight at 30 °C. The overnight culture was used to inoculate a 5 ml of appropriate selective medium 1/100 and grown shaking at 30 °C. After 2 h, β-estradiol (Sigma Aldrich) was added to a final concentration of 1 μM and cultures were grown shaking at 30 °C for a further 4 h (unless otherwise stated) before being washed in Dulbecco's phosphate-buffered saline (DPBS, Life Technologies), serially diluted and plated onto appropriate solid media without selection for pSCW11-creEBD. To determine the extent of cell death or inviability due to SCRaMbLE in a particular experiment, an equivalent culture was inoculated from the same overnight culture as the SCRaMbLE-induced culture and went through the same process but without the addition of β-estradiol. Viability is the colony count from the induced culture as a proportion of the colony count from the uninduced culture.

**Post-SCRaMbLE selection for violacein production**. Following SCRaMbLE of synV-pJCH017, colonies were picked and used to inoculate 600 μl SDO URA$^-$ cultures in a 96-deep-well plate and grown shaking at 700 r.p.m. at 30 °C. After the plates were grown to an OD$_{700}$ between 0.5 and 0.7, 10 μl of culture from each well was spotted onto SDO URA$^-$ agar and grown at 30 °C for 3 days. The darkest spotted culture was selected for further characterisation.

**96-well plate assays**. Single colonies were picked and used to inoculate 5 ml of appropriate selective glucose medium and cultured overnight at 30 °C. The OD$_{600}$ of cultures was then determined by spectrophotometry using an Eppendorf BioSpectrometer. Cultures were normalised to an OD$_{600}$ of 0.2, washed in DPBS and then resuspended in the appropriate growth medium. Normalised cultures were then used to inoculate wells in a 96-well plate to an OD$_{600}$ of 0.02, up to a final volume of 100 μl. Plates were incubated in a Biotek Synergy HT plate reader for the duration of the growth assay at 30 °C, shaking at the medium setting between readings. For growth curves, all values were normalised to the average OD$_{600}$ value of blank media wells measured at the same time as that time point prior to statistical analysis. For fluorescence values, wells were measured with an excitation wavelength of 485 nm and an emission wavelength of 528 nm. All values were normalised to the average OD$_{600}$ value of blank media wells and then to negative control cells without a plasmid (with media blank subtracted) grown in the same assay plate prior to statistical analysis. Endpoint sfGFP quantification for comparison of SCRaMbLE induction times was performed in the same manner

except with plates inoculated using 1 µl of saturated overnight 96-well 100 µl SC LEU⁻ cultures up to a final volume of 100 µl.

**Statistical analysis**. Unless otherwise stated, values given are mean average values. Significance was determined via two-sample *t*-tests with assumption of variance determined by *F*-test. Two-tail *p*-values are given with $*p \leq 0.05$, $**p \leq 0.01$, $***p \leq 0.001$ and $****p \leq 0.0001$. Replicate numbers given indicate the number of biological replicates, except for in the case of qPCR sample analysis where in each case two technical replicates were performed of each of three biological replicates.

**Violacein extraction and quantification**. Violacein extract collection was performed by adapting a previously described method[16]. Briefly, strains were grown for 2 days in 5 ml SC URA⁻, with 1 ml of culture then removed, pelleted by centrifugation and resuspended in 500 µl methanol. This suspension was heated to 95 °C for 1 min, allowed to slowly cool for 20 min and then cell debris was removed by centrifugation. Extracts were quantified for violacein content by measurement of absorbance at 577 nm using an Eppendorf BioSpectrometer. Concentration of violacein was determined using the Beer-Lambert Law, assuming the extinction coefficient at 577 nm of violacein in methanol to be $1.7 \times 10^4$ dm³ mol⁻¹ cm⁻¹[36].

**Quantitative PCR**. All qPCR reactions were performed in an MasterCycler ep RealPlex 4 (Eppendorf) using SYBR FAST Universal qPCR Master Mix (Kapa Biosystems) according to the manufacturer's instructions. DNA quantification reactions all contained 1 ng template DNA, as determined by Qubit. A calibration curve for each amplicon was constructed by quantifying known purified pJCH017 amounts at 160, 80 20, 5 and 1.25 pg, as determined by Qubit, in triplicate. To analyse qPCR data, the threshold cycle ($C_t$) values of calibration samples were used to generate calibration curves, each with an $R^2$ value of >0.97, with which sample $C_t$ values were converted to mass values. *ACT1* control amplicons were included to ensure that starting material was equal between samples. The qPCR cycle used was an initial 3 min 95 °C denaturation step followed by 40 cycles of 95 °C for 3 s and then 60 °C for 30 s. All qPCR primers were designed using the web-based Integrated DNA Technologies RealTime PCR Design Tool (http://idtdna.com/scitools/applications/realtimepcr). To assess relative plasmid copy number, *kanR* DNA was amplified using primers BB_kanR_F (ATTCCGTCAGCCAGTTTAGTC) and BB_kanR_R (ATGTCGGGCAATCAGGTG), *vioB* DNA was amplified using primers BB_VioB_F (GACTTCAACTTGCACTTGGTG) and BB_VioB_R (GCCCCTTCTCCATAATCCTAC) and *vioD* DNA was amplified using primers BB_VioD_F (ATTATCCCATGACAGGTGCC) and BB_VioD_R (TGACCAAT CGAGAAATGACCG).

**LC-MS quantification of penicillin G**. Quantification of penicillin G in culture supernatant was performed by LC-MS as developed previously, using an Agilent 1290 LC and 6550 quadropole time-of-flight mass spectrometer with electrospray ionisation and an Agilent Zorbax Extend C-18, $2.1 \times 50$ mm, 1.8 µm LC column[19]. Penicillin G (Sigma Aldrich) standards of concentration 0 ng ml⁻¹, 10 ng ml⁻¹, 100 ng ml⁻¹, 1 µg ml⁻¹ and 10 µg ml⁻¹ were used to construct a curve of best fit that was then used to quantify Penicillin G in culture supernatant samples. Samples were prepared by growing cells to an $OD_{600}$ of 0.6, in 96-well format, centrifuging and removing the supernatant, which was used as the sample.

**Post-SCRaMbLE selection for growth on xylose**. Following SCRaMbLE of synV-pJCH006, colonies were picked and used to inoculate 600 µl SX URA⁻ cultures in a 96-deep-well plate and grown shaking at 700 r.p.m. at 30 °C for 2 days. Cultures were then diluted 1/100 and used to inoculate fresh SX URA⁻ cultures in a 96-well plate up to 100 µl final volume. The plate was incubated by shaking at 700 r.p.m. at 30 °C for 5 days with $OD_{600}$ readings being taken at various points using a Biotek Synergy HT plate reader.

**Baffled flask growth curves**. Single colonies were picked and used to inoculate 5 ml of appropriate selective glucose medium and cultured overnight at 30 °C. The $OD_{600}$ of cultures was then determined by spectrophotometry. Cultures were normalised to an $OD_{600}$ of 0.2, washed in DPBS and then resuspended in the appropriate growth medium. Normalised cultures were then used to inoculate the appropriate medium in a 1-litre baffled conical flask to an $OD_{600}$ of 0.02, up to a final volume of 60 ml. Flasks were incubated at 30 °C by shaking at 250 r.p.m. At the specified time points, $OD_{600}$ of cultures was determined by spectrophotometry.

**HPLC analysis of glucose and xylose cultures**. XD4-pJCH006 and synV-pJCH006 5 ml overnight SD URA⁻ cultures were inoculated, grown overnight, washed and normalised for $OD_{600}$ as described for the baffled flask growth curves. In duplicate, cultures were used to inoculate 250 ml conical flasks containing 25 ml SC URA⁻ with 4% glucose, SX URA⁻ with 4% xylose or SCX URA⁻ with 2% glucose and 2% xylose. Cultures were incubated at 30 °C by shaking at 250 r.p.m. At 0, 5.5, 21.33, 25, 30, 45.33, 70.33 and 118.33 h time points, 1 ml samples were taken from each culture with 100 µl analysed by spectrophotometry to determine the culture $OD_{600}$ and the remaining 900 µl being frozen at −20 °C for downstream analysis. Samples were filtered through 0.45 µm membranes prior to quatification of glucose, xylose and xylitol. Quantification was performed on samples and

standards by HPLC using an UltiMate 3000 (Dionex-Thermo Fisher Scientific) with an Animex HPX87H column eluted with 0.01 N $H_2SO_4$ at room temperature with a 0.6 ml min⁻¹ flow rate[37].

**Microscopy**. Five millilitres of cultures were grown overnight and visualised on an Eclipse Ti inverted microscope (Nikon) at 90× magnification. Images were captured using the NIS-Elements Microscope Imaging Software (Nikon). The area of a minimum of 24 cells was determined for each strain and condition tested using ImageJ (http://imagej.net).

**Nanopore sequencing**. Genomic DNA was sheared to 20 kb using a g-TUBE (Covaris) and then underwent library preparation using a Ligation Sequencing Kit 1D R9.4 (Oxford Nanopore Technologies). Each library was analysed on an R9.4 flow cell using a MinION Mk 1B (Oxford Nanopore Technologies). A standard 48 h sequencing run was performed using the MinKnow 1.5.5 software using local basecalling.

**Sequencing analysis**. Raw fast5 files were converted to fastq and fasta using Poretools[38]. Raw reads were corrected using Canu (v1.5) (www.canu.readthedocs.io). Owing to erroneous ultra-long reads for XD4, reads >60 kb and <1 kb were discarded from this corrected pool using a custom python script in combination with samtools (www.samtools.sourceforge.net), bedtools (www.bedtools.readthedocs.io), last (maf-convert function, www.last.cbrc.jp) and the fastx-toolkit (www.hannonlab.cshl.edu). For VB2, all raw reads were used for assembly. Smartdenovo (www.github.com/ruanjue/smartdenovo) was run on corrected reads to de novo assemble a contiguous sequence (contig) using default flags. This contig was compared to synV using lastal (-l100 flag) (www.last.cbrc.jp) and viewed using ACT (www.sanger.ac.uk). Corrected reads were also aligned to synV using lastal (flag –l100) and viewed on integrative genome viewer (IGV) (www.software.broadinstitute.org) to verify SCRaMbLE events seen in the assembled contig.

**PCR verification of SCRaMbLE events**. Genomic loci identified as potentially containing SCRaMbLE events were amplified from genomic DNA by PCR using Phusion Hi-Fidelity DNA Polymerase (New England Biolabs). The VB2 deletion of UBP3 was confirmed by amplification with primers GG047 (TCCTGAGATAAA GTATGGTGCTTCG), GG048 (CTCGCCAGACAGCTATCTAAAGG), GG049 (GAATCAGAAGCTTCTTCGCCAC) and GG050 (GTCAGACTCGTCTGCT ACCATC). The VB2 inversion event was confirmed by amplification with primers GG051 (GCGTGCAAAGTTTGCTCAAGTTGAC), GG052 (TCAAAACCGCTTT CGCAGCAGATC), GG054 (CCAAGTTCAACCCGTTGGTTTACTGC) and GG075 (GCAGTAAAGCCGTAAAATTGAACG). The XD4 *MXR1* deletion was confirmed by amplification with primers GG035 (TTGACGTTTGGAAGGACGT AATAG), GG036 (TTCAAGGCCGACCACTACAG), GG037 (GCCACCACAG TTGATCGC) and GG038 (TCTCAGATAATGAGTAGGGCATGC). The XD4 inversion of the *GCN4-YEA6* region was confirmed by amplification with primers GG031 (TGGGCGAATAGCAGAGCTG), GG032 (GATGACGAATCGAGAC TGGATC), GG033 (GTGTGCACTGCCGAAATAGC) and GG034 (GAACACG ATGAATTACCAGCAGAG). PCR products were separated by agarose gel electrophoresis, stained with Sybr Safe DNA Gel Stain (Invitrogen) and visualised with a Gel Doc XR transilluminator (Bio-Rad).

**CRISPR-mediated recombination**. SCRaMbLE deletion regions of VB2 and XD4 were amplified from gDNA prepared from those strains by PCR using primer pairs BB632 (GCCCTAGGTTTGGCTGG)/BB633 (GGTGAGGACTCGTTCGG) and BB548 (ACCACAGAAGAATAACTTGGATGAG)/BB549 (ACCGTTACTTGTT GTCTTCTGGTAG), respectively. This template DNA was co-transformed into BY4741 along with linear fragments encoding Cas9 and a gRNA retargeted to either *UBP3* (CTTAACGCAATCGAACCCAG) or *MXR1* (CCACAAACTCTC TTATAAGA) as appropriate, following the protocol described at http://benchling.com/pub/ellis-crispr-tools.

**Data availability**. The authors declare that the data supporting the findings of this study are available from the corresponding author on request.

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

## Acknowledgements

We thank Leslie Mitchell and Jef Boeke for plasmid pJC104, David Bell and Ali Awan for assistance with penicillin yield measurements, Young-Kyoung Park and Soo Me Chee for assistance with HPLC measurements, John Heap for advice on pathway redox and Sujai Kumar and Henry Lee for discussions regarding nanopore sequencing. This work was funded in the UK by BBSRC awards BB/K019791/1 and BB/P504579/1 and funded in China by the Ministry of Science and Technology of China (2014CB745100 and 2015DFA00960) and the National Natural Science Foundation of China (21390203 and 21621004).

## Author contributions

B.A.B. and T.E. conceived and designed the experiments, B.A.B., G-O.F.G., J.C.H.H., R.L.-A. and D.J. performed the experiments, B.A.B., J.C.H.H., G-O.F.G. and R.L.-A. analysed the experimental data, G-O.F.G., R.M.McK and T.E. analysed the sequencing data, Z.X.X., B.Z.L. and Y.J.Y. generated the synV strain, B.A.B., G-O.F.G. and T.E. prepared the manuscript.
