## [Peer Review File · Nature Communications]

We thank the reviewers for their constructive comments and questions. The original reviews for [REDACTED] were particularly helpful in highlighting that the manuscript would be improved by greater description of the approach taken and a more thorough justification of the SCRaMbLE method as a promising addition to the various existing strain improvement tools used in yeast. Therefore, we have now significantly altered the structure and content in the revised manuscript. The increased word limit of the Nature Communications article format has enabled us to elaborate on new several points and better frame this work in the context of existing studies and techniques. We narrow the scope of the paper to be a description and demonstration of the use of SCRaMbLE as a host strain improvement tool. In response to reviewer criticism, we have now moved speculation as to the genetic basis of strain improvement into the discussion. Studies that fully-determine the link between rearranged chromosomes and their phenotypes will require substantial extra experimental work and so we feel are beyond the scope of this initial paper.

Having said that, we have performed several additional experiments in response to specific points raised by the reviews. These include work to (i) better characterise the SCRaMbLE method and the frequency with which it improves phenotypes, (ii) discovery that the improved XD4 phenotype was due to that strain bypassing a xylitol accumulation observed in the parental strain, and (iii) full genome sequencing of the VB2 strain to reveal the SCRaMbLE events that have occurred. We are confident that these, and other additional changes have improved the scientific content of the manuscript, better explain the reasoning, context and benefits of the technique and address any concerns previously raised by the reviewers.

Finally, we would just like to briefly apologise for the length of time it has taken to produce this revision. While generating the substantial new data presented in the revised manuscript took several weeks, our delay is primarily due to two factors; (i) many repeated and ultimately failed attempts to use genome engineering to recreate a SCRaMbLE inversion event in a non-synthetic background in order to address a reviewer request, and (ii) the need for our team to also prioritise our work to finalise production of synthetic yeast chromosome XI for the Sc2.0 project in time for 2018.

Reviewer #1:

Remarks to the Author:

Summary:

The manuscript by Blount et al describes the development of a platform for strain engineering that leverages modular genome re-arrangements in yeast with a synthetic chromosome. They demonstrate the ability to induce combinatorial rearrangements of the genome of SynV by SCRaMbLE, a previously established technique, to improve host strains for violacein synthesis, penicillin synthesis, and xylose utilization. This work has applications for synthetic biology, metabolic engineering and genome engineering. In general, this manuscript is well written, the data are clearly presented, and the study is performed at a high level. Despite this, I have some concerns over the possible impact and originality of the work and whether or not it should be published in a more specialized journal in its current form, despite the fact what is presented is well done.

We thank the reviewer for their positive appraisal of the quality of the manuscript and experimental work. We agree that the framing of the work within the context of existing studies and techniques required improvement and thus we have now significantly altered the manuscript, adjusting the introduction, adding new data and expanding the discussion in order to better highlight the originality of the approach we demonstrated and its impact compared to other methods.

Our work is highly original in that it represents the first time that the emerging technique of SCRaMbLE has been demonstrated as a strain improvement technique for enhancing targeted phenotypes, the first time that SCRaMbLE has been combined with the expression of heterologous pathways and phenotypes, the first time that specific full chromosome-scale SCRaMbLE events have been identified and the first time that nanopore sequencing technology has been used to identify SCRaMbLE events. These multiple demonstrations in combination represent a proof of principle that SCRaMbLE allows improvement of predetermined heterologously encoded phenotypes and would be a valuable addition to the existing molecular biology, metabolic engineering and strain improvement tools. The technique has several qualities that compare favourably to existing techniques, including the speed in which improved strains can be isolated and characterised, the lack of requirement for specialised equipment or techniques, the low associated cost and the extremely large potential genetic space sampled. These are now highlighted in an added table that compares commonly-used yeast strain improvement approaches (**Table 1**).

Major concerns:

The key innovation seems to have already been reported. At the core of this work is the SCRaMbLE technology, which has already been published (Shen et al. Genome Research 2015 genome.cshlp.org/content/26/1/36; Jovicevic et al. Bioessays 2014 onlinelibrary.wiley.com/doi/10.1002/bies.201400086). I agree with the authors that SCRaMbLE provides a new form of genome diversification. I also think that the authors apply SCRaMbLE to generating host strains for metabolic engineering for the first time to my knowledge, but the innovation seems to be SCRaMbLE. A challenge is that the whole genome is not targeted as I understand it.

We respectfully disagree with the reviewer's assertion that the key innovation of the paper is the SCRaMbLE system itself. The advancement offered in this work is in the use of the system as a powerful strain improvement tool for the optimisation of multiple heterologous pathways in yeast.

A demonstration of SCRaMbLE was first described in Dymond et al (doi:10.1038/nature10403) and, subsequently, Shen et al (doi:10.1101/gr.193433.115) applied sequencing technologies to strains containing SCRaMbLEd iterations of a circular 100 kb synthetic chromosomal arm to identify what rearrangements had occurred. We feel that our work to use SCRaMbLE as a tool to improve specific, predetermined engineered phenotypes is notably different to these studies which confirm SCRaMbLE events occurring, but do not demonstrate that SCRaMbLE can give strain improvement.

Although the whole genome is not targeted in the work presented, the synthetic chromosome V SCRaMbLEd in this work is over 5 times larger than that SCRaMbLEd in the work by Shen et al and is sufficient to produce the enhanced phenotypes described. We agree that SCRaMbLEing a whole genome would increase the available genetic space, and once the entire synthetic yeast chromosome is completed (est. 2020) we aim to perform these experiments. The space sampled by SCRaMbLEing this one chromosome with 174 individual symmetrical recombination sites is, however, extremely large and we have demonstrated that the resulting diversity is sufficient to isolate the improved strains sought in our example experiments.

In the revised manuscript we have now adjusted the introduction text to clarify the advancement that this work makes on previous SCRaMbLE studies (**Lines 50-56**).

I didn't find any of the results especially surprising. For example, SCRaMbLE has been used before and there have been past efforts on using genome engineering tools to develop host strains. If the authors would like to

establish their method as better than the state of the art, I believe they need to compare it to such methods and show that it is better by some defined metrics (cost, speed, diversification, etc.). For example, how does this technique compare to other established genome engineering strategies to engineer host strains, such as transposon mutagenesis or global transcriptional machinery engineering, among others? As presented, I can't evaluate the differences. Also, it was striking that multiple rounds of SCRaMbLE were not beneficial. This is a concern for adoption by others and may limit design space.

We have found the technique to be very powerful, and we agree that improved comparisons to existing methods would benefit the manuscript. Thus we have made appropriate alterations to the introduction and discussion text and have added **Table 1** to compare SCRaMbLE to establish strain diversification methods in order to address this. The SCRaMbLE technique compares very favourably to existing techniques in many criteria, particularly as an initial method that is a much faster and low-cost exploration of potential genotypic/phenotypic space when compared to the other examples cited. The hard work in implementing a SCRaMbLE system is in the building of synthetic chromosomes and incorporating the loxPsym sites as a design change. As these chromosomes are being built by the Sc2.0 consortium and will be freely available to researchers and to companies upon completion, anyone wanting to use SCRaMbLE to improve heterologous phenotypes will be able to benefit from the system having already been engineered into the genome by the consortium.

Additionally, the new experimental work presented in the revised manuscript on the diversity of sfGFP expression generated by SCRaMbLE (and the frequency at which improved colonies are generated) further demonstrates the power of the method. We also show that longer SCRaMbLE periods do not have the same detrimental effects as those seen in the re-SCRaMbLEing experiment. This new work is described in **Lines 265-314** and forms a new figure (**Figure 4**).

I was disappointed that the pathways used (violacein, penicillin G, and xylose utilization) have all been published before. I do applaud that the authors made penicillin G, which is not coloured, but they only make it in a few select strains. If the key advance is that this is a rapid approach for metabolic engineering, it can only be rapid and generalizable if a screening technology matches the genome engineering speed.

The thinking behind the use of violacein, penicillin G and xylose utilisation as target pathways is that they are a varied demonstration of the types of phenotypes that could be enhanced. Violacein production allowed us to look at a relatively well-studied pathway which has also been optimised using a different method (Lee et al doi: 10.1093/nar/gkt809) and so can be directly compared to these efforts. Xylose utilisation represents an entirely different type of pathway optimisation to production of a product biomolecule. Penicillin G represents an extremely difficult pathway to express in *S. cerevisiae* that has only recently been reported (Awan et al doi:10.1038/ncomms15202), and the SCRaMbLE-enhanced strain described here produces double the penicillin levels of the previously best-producing strain. As the manuscript is intended to describe a method that we envisage to have broad utility, we feel that addressing these multiple different themes was a relevant demonstration of the strengths of the technique. As the reviewer notes, a grand challenge of the field currently is to develop screening methods that match the power of strain diversification methods.

The current limitations in screening have directly influenced our screening strategy in the manuscript. The XD4 and VB2 strains were intentionally picked and screened without any prior selection in order to demonstrate that improved phenotypes can be isolated using any basic screen that is amenable

to 96-well plate throughput. Of course, as new high throughput screening techniques emerge, the throughput of analysis of SCRaMbLEd strains will improve and this will only help, rather than hinder, the technique. The revised manuscript now contains a new results section (**Lines 265-314** and **Figure 4**) that specifically addressed questions on screening strategies and the merits of random and selective selection.

Technical questions:

In Figure 1 what are the units of penicillin G produced? What is the concentration made? There is also no comment on whether this 2-fold increase or the 2-fold increase in violacein obtains an industrially-relevant or high amount relative to other publications on these molecules (which many exist). Usually a 2-fold increase is not sufficient for metabolic engineering applications. What is the concentration made?

The units in Figure 1 were given as LCMS counts to indicate relative values between VB2 and the controls. The graph text has been changed to convert these values to quantitative amounts of penicillin (i.e. ng/ml). The VB2 strain produces 14.9 ng/ml, with the previous highest published yield being ~5 ng/ml (Awan et al doi:10.1038/ncomms15202). Results text is updated in **Lines 123-124**

Also, can the authors comment on what may have caused the increase in copy number and if there might have been other genomic changes leading to enhanced yields?

We agree that the lack of understanding of the genomic changes responsible for the VB2 phenotype was a weakness of the manuscript. We have now fully-sequenced VB2 and identified the two specific SCRaMbLE events in that strain, a deletion of UBP3 and an inversion of the SWI4/LSM4 locus. **This work has now been added to Figure 3.** No other major changes to the genome were evident in the full genome sequencing of VB2, and so in the revised discussion section we speculate on the link between the identified SCRaMbLE events and the resulting phenotype, noting that changes to SWI4 expression are likely to affect the cell cycle and DNA replication (**Lines 374-381**).

In Figure 2, the authors claim that a second round of SCRaMbLE is detrimental. Would a second initial round of SCRaMbLE produce different, potentially more beneficial, results?

In the new results section concerning SCRaMbLE induction times and screening, we now show new experimental data that demonstrates that longer SCRaMbLE periods (e.g. 8 hours) indeed do produce beneficial results compared to shorter SCRaMbLE periods (albeit for only one desired phenotype). Note that a second round of SCRaMbLE without intervening selection is effectively just a longer SCRaMbLE period, and this is also now mentioned in the new results section (**Lines 265-314** and **Figure 4**).

In Figure 2, there are no graphs showing how good the strains are at utilizing xylose. An additional figure or supplemental data needs to be included to show xylose concentration over time in shake flasks in order to compare this work to the state of the art. The authors claim that their strains are better at xylose utilization than the state of the art, but it is unclear to me that this is the case without these data.

We thank the reviewer for this comment, and agree that the manuscript greatly benefits from this characterisation work. The suggested experiment has been carried out and provides direct evidence for, and allows us to elaborate on, our initial claims regarding xylose utilisation of XD4. **This new experiment has been added to Figure 2** and is described in **Lines 183-191** of the results section. Slow growth of synV in xylose is due to an accumulation of xylitol, symptomatic of a cofactor imbalance affecting the xylitol dehydrogenase-mediated step of the heterologous xylose metabolism pathway. This accumulation is not evident in XD4, indicating that the imbalance has been ameliorated in the SCRaMbLEd strain.

In Figure 2, what changes were responsible for the improved phenotype?

As mentioned above, our new HPLC data on xylose consumption and xylitol accumulation indicates that the phenotype of improved growth on xylose is due to a redox change in the cell. In the new discussion section, we briefly speculate as to how the deletion and inversion SCRaMbLE events identified by sequencing in Figure 3 could explain this (**Lines 381-388**).

The authors in Figure 3 describe a sequencing effort to read the whole SCRaMbLEd chromosome. However their results are inconclusive as to why the changes that appeared were beneficial for the strain enhancement, with the authors essentially concluding that the results are complex. I think the authors need to sort out the biological mechanism for their response.

Continuing from above, in the new discussion section (**Lines 381-388**) we speculate on the changes that lead to the phenotype for XD4 (and also for VB2). For XD4, the deletion of *MXR1* may indirectly change the redox state of the cell, but the observed phenotype may also rely in part to the inversion event which changes the 3'UTR sequence for *GCN4*, a metabolism master regulator gene known to have post-transcriptional expression control. As described in the discussion section (**Lines 389-391**), further work would be required - ideally with proteome and metabolome experiments - in order to fully-map the phenotype to the genotype. We feel that experimental studies down this route are beyond the scope of this manuscript, which is intended to introduce using SCRaMbLE for strain diversification to improve heterologous pathway performance, and not to determine new molecular mechanisms for global metabolic changes.

Additionally, I would like to see metabolomic profiles for their enhanced strain. If their hypothesis was incorrect as to why the strain was enhanced what else is happening in the strain?

While such further work would indeed be interesting, the purpose of our manuscript is to describe the SCRaMbLE method as a broad tool for rapid strain diversification that can improve the performance of different heterologous pathways. We feel that an in-depth metabolomics effort to understand the underlying mechanisms of specific phenotypes (such as xylose utilisation) is beyond the scope of this paper. As it stands, we feel that the additional characterisation of xylose utilisation by XD4 at least provides further information on this enhanced strain. We agree that further in-depth metabolomic studies of enhanced strains for specific applications would, of course, represent a rich area of investigation.

Reviewer #2:

Remarks to the Author:

Summary:

In this paper, the authors demonstrate the utility of SCRaMbLE to generate optimized metabolic pathways using a yeast strain carrying a synthetic chromosome. In general, this was a previously published technique in which the synthetic yeast strain rearranges itself for a short period of time upon exposure to Cre recombinase. The authors initially demonstrate enhanced (~2-fold) violacein production from a 2 μ plasmid. The authors attribute the increased production due to increased plasmid copy number. The authors demonstrated this principle with several additional examples, including optimized growth in specific media, which has industrial applications for metabolic engineering. The authors use long-read nanopore sequencing to infer mechanistic insights from chromosomal rearrangements, such as reduced oxidative stress response from deletion of MXR1. However, they note that the inverted 7 kb locus may also be a contributing factor, and overall changes could be a result of one or both changes.

General comments

Essentially, the work represents a combination of three recently established technologies (synthetic yeast chromosome, SCRaMbLE, and nanopore sequencing) to achieve optimisation of metabolic pathways. The integration led to the successful improvement of the example pathways. While the work has potential, what is missing is a clear demonstration of technical or conceptual advance; the presented results are incremental.

We thank the reviewer for their constructive comments, questions and suggestions. We agree that the context of this work could have been better explained and now in the revised manuscript we have adjusted the introduction (**Lines 50-56**) to clarify the advancement that this work makes on the previously published study of SCRaMbLE.

We are confident that together the work we present is more than just incremental work. Previously, Shen et al (doi:10.1101/gr.193433.115) applied sequencing technologies to strains containing SCRaMbLEd iterations of a circular 100 kb synthetic chromosomal arm to identify what rearrangements had occurred. However, an application of SCRaMbLE has not been previously shown. Our work is a significant advancement as it (i) represents the first time that the emerging technique of SCRaMbLE has been demonstrated as a strain improvement technique for enhancing engineered phenotypes, (ii) the first time that SCRaMbLE has been combined with the expression of heterologous pathways and phenotypes, (iii) the first time that specific full chromosome-scale SCRaMbLE events have been identified and (iv) the first time that nanopore sequencing technology has been used to identify SCRaMbLE events. These multiple demonstrations in combination represent a technological proof of principle that SCRaMbLE allows improvement of predetermined heterologously encoded phenotypes. This, in our view, is a clear advance on previous work, and shows that SCRaMbLE represents a viable and broadly-applicable strain improvement tool. Thus it is of broad interest and wide utility.

A number of important questions are not addressed. What is the advantage of using a synthetic chromosome? If indeed there's an advantage in using a synthetic chromosome, what features of the synthetic chromosome give this advantage? Is the purpose of the work to demonstrate this advantage or that of SCRaMbLE? It is not

surprising that Cre-Lox is capable of producing genotypic variety, with some variants offering desired phenotypic improvements. Or is it to demonstrate the use of nanopore sequencing to learn something about the scrambled pathway? The authors alluded to the last point but as detailed below, few mechanistic insights were presented.

In the revised manuscript we have now altered the text in the introduction section to better clarify the aim of the research we describe in the manuscript (**Lines 57-67**). The purpose of the work is to demonstrate SCRaMbLE as a new tool for rapid strain diversification, and show how it can improve the performance of different heterologous pathways. The manuscript is intended to describe a methodological proof-of-concept using multiple pathways as examples and further show that nanopore sequencing offers a tractable way to identify the SCRaMbLE events. Going beyond this to use the method to gain complete mechanistic insights into specific phenotypes is beyond the scope of this initial paper, but will hopefully be soon presented in a future work (pending a funding decision).

The use of a synthetic chromosome is a fundamental requirement in this work as the SCRaMbLE system is enabled by the incorporation of 174 loxP sites throughout this one chromosome alone. It currently is not possible to separate the two concepts. Other techniques, such as MAGE and its derivatives, could theoretically replicate this degree of change throughout a genome, but to do so would be an extremely lengthy process. Using nanopore sequencing to gain insights into strains that have been enhanced by SCRaMbLE is very much relevant here as it demonstrates that the chromosomes have indeed rearranged at specific loxP sites.

We agree that the results section in initial manuscript was weak in exploring the link between genotype and phenotype. To address this, we have now provided more experimental data in the form of full sequencing of the VB2 genome (**see updated Figure 3**) and HPLC analysis of XD4 cultures that reveals that the strain bypasses xylitol accumulation (**see new results in Figure 2**). Going beyond this, to map how the identified gene rearrangements lead to the global changes in metabolism, is likely to be a major undertaking, and so we now move speculation on why SCRaMbLE events have led to the observed phenotypes into the new discussion section (**Lines 366-400**).

Specific technical points:

1. The authors claim short SCRaMbLE to be optimal, and that 4 hours achieved this. Next, they demonstrate that a second round of SCRaMbLE will over-rearrange the chromosome resulting in unwanted effects on the strain. In general, this raises several questions:

Is 4 hours optimal? Did the authors do the same experiment at incremental hourly time points of SCRaMbLE lengths to show this? What does this optimality look like – is it linear until a threshold is met or is it gradually biphasic etc.? Ideally these are important parameters to quantify and should be elaborated upon. Is two rounds of SCRaMbLE at four hours each equal to one round of SCRaMbLE at 8 hours (i.e. is the length of time scrambling linearly correlated with rearrangements or is there a lag period or some other explanation this wouldn't be an additive process)? Is this true regardless of the time window (e.g. two rounds of SCRaMbLE for two hours equal to one round at four hours)?

We thank the reviewer for these important questions. In order to address these requests for better characterisation of the SCRaMbLE process itself, we have now carried out more extensive and quantitative experiments SCRaMbLEing synV with an sfGFP-expressing plasmid. We have added this work as a new results section (**Lines 265-314** and **Figure 4**). Although it can be assumed that dynamics of the optimisation of hosts will vary from pathway to pathway (and from synthetic

chromosome to synthetic chromosome) we feel that this is an appropriate way to broadly investigate appropriate SCRaMbLE induction times and expected frequencies of improved performers. These experiments indicate that with random sampling, an 8 hour induction period generates the highest proportion of colonies with higher sfGFP production than the controls (23.5%), whereas a 24 hour induction produces the highest performing individual colonies. This is also true with selective sampling although, as expected, the proportion of colonies with improved expression is much higher (94.7% for the 8 hour induction). In terms of whether 2 rounds of induction at 2 hours is equivalent to a single 4 hour induction, we would expect this not to be the case. As estradiol is not removed from the cells themselves, only removed from the media, cells in which estradiol is bound to Cre will continue to be susceptible to SCRaMbLE until the recombinase is degraded. This lag period would be effectively doubled in the case of 2 rounds of 2 hours, compared to a single 4 hour induction.

Based on the way the paper is written, it sounds like a second round of SCRaMbLE simply over-scrambles the chromosomes, and therefore it should not be claimed that two rounds was deleterious, but instead, that long SCRaMbLE is the true culprit.

Our additional characterisation of SCRaMbLE (**Lines 265-314** and **Figure 4**) has now shown that longer SCRaMbLE times did not produce notably fewer cells with enhanced performance (at least for the example of improved expression of sfGFP). Thus it does seem that the second round of SCRaMbLE was deleterious in the xylose example discussed in the original manuscript. We assume that the deleterious effects of two rounds of SCRaMbLE when interspersed with a screening phase arise from the screening and isolation of just one SCRaMbLE derivative. This then severely limits the available phenotypic space for the second round of SCRaMbLE. This would perhaps be analogous to a species being committed to a specific evolutionary branch and, in doing so, losing the ability to explore additional, but mechanistically unrelated, potential ways of improving a phenotype. This thinking is now summarised in **Lines 308-314**.

This raises an additional important point, which is whether multiple parallel SCRaMbLE rounds for the same pathway result in the same or a small subset of reproducible genomic alterations? In other words, how reproducible are the changes made resulting in significant changes for a particular product?

This is an excellent question. To address this, we took colonies that showed enhanced sfGFP expression from a 2 micron vector from the experiment described above and screened these for the SCRaMbLE events identified in VB2. As none of the strains were found to have a UBP3 deletion or a SWI4/LSM4 inversion this indicated that different SCRaMbLE events were responsible for the enhanced phenotypes. While intriguing, this is not surprising as the potential combinatorial genotypic space arising from the SCRaMbLE of a chromosome with 174 symmetrical recombination sites is huge, and so in most cases we would expect different “solutions” to phenotype enhancement to arise. This work is now described in **Lines 275-301** in the new results section.

2. The conclusion that increased sfGFP expression confirms that the plasmid copy number has doubled needs significantly more direct evidence. That two different plasmids demonstrated increased expression is not a proxy for a measurement of copy number. In fact, many other explanations can explain this observation, such as decreased activity of protein degradation enzymes (or stability) within the cell, enhanced activity of key transcription proteins, not to mention changes in key metabolic genes that regulate the intracellular

environment, which may increase the accumulation of either transcripts or proteins, etc., that would affect both plasmids.

We provide direct evidence for an increase in plasmid copy number via the quantitative PCR determination of relative plasmid copy numbers targeting 3 different sites on the plasmid (**see Figure 1f**). This data indicates that plasmid copy number has increased to a high degree of statistical significance ($p \leq 0.0001$).

Critically, and especially when compared to penicillin G production, the authors say nothing of the comparable biomass, which could account for a two-fold increase if the growth rate was also increased.

The penicillin data have been normalised to OD_{600} (as a proxy for biomass) in order to account for any differences in this respect. For further clarity, an additional supplementary figure, **Figure S7**, has been added to demonstrate that the effects of this normalisation are nominal.

Minimally, the authors should provide more direct evidence for increased plasmid copy number, as well as sequence evidence for the violacein plasmid to demonstrate no mutations occurred.

As mentioned above, we have supplied direct evidence for differences in plasmid copy number in VB2 via quantitative PCR measurement (**Figure 1f**). As we show that the increased expression of pathways from 2 micron plasmids was a phenotype observed across 4 different plasmids that were expressed in VB2, we feel it is reasonable to discount that the phenotype is the result of a mutation on one of these four plasmids. We have modified the results text (**Lines 93-95**) to add that we saw the same enhancement phenotype when we retransformed VB2 with the original plasmid stock.

3. In general, there is only mentioning of the optimized pathway and no mentioning of the potential detrimental effects this pathway might have on overall cell physiology or viability of the cell. This point should be elaborated on. For example, if the purpose is for long-term bioreactor culturing for continuous production of key proteins off the 2 μ plasmid, simply demonstrating enhanced expression or production of the protein does not in itself demonstrate optimized functionality. Instead, the authors need to show that the other critical dynamics are working within a similarly desired range as before the rearrangement. For example, increased optimization might actually lead to significantly shorter lifespan in which case the pathway hasn't truly been optimized if the goal is to use it for a long-term application.

The revised manuscript has altered the text in the introduction and discussions sections to better clarify the aims of the research described in the manuscript. We hope that our rewording of the text has made it clear that the intent of this work is to describe SCRaMbLE as a tool for strain diversification that can improve expression of heterologously expressed pathways. It is certainly not the intent of this work to show that SCRaMbLE can generate a specific industrial bioproduction strain. We do believe that strains generated by this method would be of interest for commercial bioproduction but agree with the reviewer that for those purposes they would need to undergo additional and extensive testing in order to determine their applicability for industrial scale-up. We have now added text to the discussion to mention this (**Lines 396-400**). As the initial synV strain was shown to have no significant growth defects or longevity issues compared to BY4741 yeast (Xie et al, Science 2017 doi:10.1126/science.aaf4704), we would only expect to see such phenotypic defects arise from SCRaMbLE events.

4. Although the authors claim novelty in the sequencing aspects as well, they only speculate on the sequencing results, which generate no definitive conclusions. This significantly reduces the utility of sequencing for understanding rearranged chromosomes. In particular, knowing the particular rearrangements, the authors should re-engineer strains with each of the identified discrepancies identified (e.g. the inverted 7 kb region or deleted 785 kb region), and test whether they can fully or partially explain the observations. This would show the utility of nanopore sequencing, and how it was beneficial. I think this point needs to be critically expanded upon.

We apologise that the initial draft of the manuscript did not properly describe the main purpose of the manuscript (*i.e.* to demonstrate SCRaMbLE as a technique to generate isolates with improved phenotypes) and we hope that our revisions now make the aims and scope of this work clearer. The purpose of the sequencing in this work is to (i) show that SCRaMbLE events have occurred in the strains and (ii) demonstrate a workflow where the specific events can be identified using nanopore sequencing. To this end, the sequencing of strains VB2 and XD4 (**Figure 3**) have enabled us to identify the recombination events in each strain which, in itself, demonstrates that the sequencing method is successful.

In terms of re-engineering the strains with the discrepancies identified, upon receiving this comment we set out to do precisely this for the XD4 strain. While the deletion was trivial (and led to improved growth on xylose), recreating the inversion region proved impossible for us to achieve via DNA cloning and CRISPR-mediated insertion/deletion, partly due to a very repetitive, very high-AT region within that could not be amplified by PCR or synthesised on time by commercial DNA synthesis companies. In fact efforts to recreate the inversion consumed much of the several months since our manuscript was returned to us. While this is disappointing, it does highlight that SCRaMbLE can generate chromosome changes that would be very difficult to achieve via any other methods. However, without achieving this we now prefer to move our speculation on the molecular mechanisms causing the XD4 phenotype to the discussion section (**Lines 374-391**) and indicate further experiments that can be done in future studies for determining the link between the chromosome rearrangements and the altered metabolism.

Having said that, during the revision period we also now characterised both XD4 and VB2 strains further, gathering additional data. We have shown that XD4 has improved xylose utilisation by overcoming a xylitol accumulation in the parental strain (shown in previous work to be due to a redox cofactor imbalance) and that VB2 cells have a larger size than normal cells when grown in rich media. These new data give further clues to help map the link between observed genotypic rearrangements and the phenotypes.

5. What happens if the authors repeat the process with the same selective levels -- will the same alterations in the genome be identified? This would suggest a single optimization pathway that is favored over others, potentially contributing to my previous point that optimization also minimizes detrimental rearrangements. At least one experimental repeat and analysis should be performed.

An excellent question. To address this, we have now performed additional SCRaMbLE experiments with varying parameters, which are now added to the new results section (see **Figure 4**). Colonies that showed enhanced sfGFP expression from a 2 micron vector post-SCRaMbLE were isolated and screened for the SCRaMbLE events originally identified in VB2. As none of these strains were found to have a UBP3 deletion or a SWI4/LSM4 inversion, this indicates that different SCRaMbLE events

were responsible for the same enhanced phenotypes. This is not surprising, as the potential genotypic space arising from the SCRaMbLE of a chromosome with 174 symmetrical recombination sites is huge and so in most cases we would expect different “solutions” to phenotype enhancement to arise. This is discussed in **Lines 299-301**.

6. The idea development, that there is a difference between a specifically engineered pathway, and an engineered strain to enhance the said pathway, should be further developed. As it stands now, the distinction drawn by the authors is unclear.

We thank the reviewer for this comment. In the revised manuscript introduction, we now clarify the text at the start of introduction in order to make clearer the distinction between direct engineering of the genes a pathway (e.g. via codon optimisation, tuning of promoter strength and combinatorial library screening) and the development or selection of a host for a pathway via strain selection or modulation of host gene expression (**Lines 32-39**). This includes references to several papers and review articles (references 4 to 8) that discuss this well-established idea.

7. The authors should elaborate on the design of the protocol. Why only 87 colonies, what is the throughput and how does this scale? Does having only one viable option from 87 clones prevent scalability or is this sufficient for other, larger pathways with more genes or involved pathways? Because these are larger segments being moved, have the authors done any characterization on the mathematics behind this probability level?

We agree that the reasoning behind the screening process should have been further explained and so we have now added further description to the manuscript text in the new results section in order to clarify this (**Lines 266-270**). We simply aimed to replicate what we believe a single round of 96-well scale screening would look like, and minus wells for controls, that led to 87 colonies. Although higher throughput assays would be available for many application examples, we felt that the 96-well scale represents a reasonable lower bound of expected screening throughput and therefore demonstrates the broad applicability of the method we demonstrate here.

Reviewer #3:

Remarks to the Author:

SCRaMbLE (Synthetic Chromosome Recombination and Modification by LoxP-Mediated Evolution) is one of the promising synthetic biology tools with which combinatorial rearrangement of the yeast genome can be induced. This is the first report to apply SCRaMbLE for the rapid and practical improvement of yeast host strain. They presented two successful examples in this manuscript. One is to develop a new yeast strain capable of doubling the expression of genes placed on 2-micron plasmids. The other one is to develop a yeast stain specialized for xylose utilization which has a major focus in the engineered yeast. However, they failed to show the compelling evidence to understand the molecular mechanism of the improvement. As for the first case, the authors did not determine the post-SCRaMbLEd chromosomes and therefore had no idea about the mechanisms of two-fold increase in copy number of the 2-micron plasmid.

We thank the reviewer for their constructive comments, questions and suggestions. We apologise that the initial draft of the manuscript did not clearly describe the intended scope of the manuscript. We have now revised the text to clarify that the manuscript aim is to demonstrate SCRaMbLE as a new technique to rapidly generate isolates with improved phenotypes (see **Lines 57-70**). While it would be interesting, we feel that fully determining and understanding the molecular mechanisms at play in the strains generated here is beyond the scope of this paper. Indeed, one would expect such work to be a study in and of itself, especially given that it is well-established that the links between genotype and phenotype for metabolic systems are often highly non-linear, meaning that full identification of mechanisms would likely require multiple layers of 'omics' experiments (see recent review by Haas *et al* <https://doi.org/10.1016/j.coisb.2017.08.009>). We have now added a section to the discussion discussing routes for future work towards determining the molecular mechanisms involved, once users of the SCRaMbLE approach have isolated improved their strains (see **Lines 389-391**).

While not fully-elucidating the molecular mechanisms, we have also now added significant further experimental data which are presented throughout the revised manuscript and the supplementary materials. These experiments expand our characterisation of VB2 by fully sequencing the genome to reveal the SCRaMbLE events (see **Figure 3**), and add further exploration of the XD4 phenotype via HPLC analysis of metabolite consumption and accumulation (see **Figure 2**).

Regarding the second example, they identified two chromosome sites for arrangement but did not perform any further experiments to explore the mechanism.

As mentioned above, a complete understanding of the molecular mechanisms is beyond the scope of this work. As such speculation on the link between the identified rearrangements and the phenotypes has now been moved to the discussion section. However our additional experimental work, now provides more insight into the observed phenotypes. This new work (see **Figure 2**) shows that XD4 has improved xylose utilisation by overcoming a xylitol accumulation in the parental strain. This is detrimental accumulation relates to a known redox cofactor imbalance in this pathway. Thus, the new data supports the hypothesis of strain improvement via redox rebalancing.

*Recent detailed study of post-SCRaMbLE strains clearly showed that the synthetic system functions as designed (Shen *et al.*, *Genome Research* 2016). Therefore, we have already known that this system is valuable in combinatorial exploration of genomic diversity for phenotype-based selection. While Shen *et al.* (2016) analyzed 64 SCRaMbLEd strains by deep sequencing, only one improved strain was characterized by analysis of genome organization in this study.*

We respectfully disagree with the reviewer regarding the content and findings of the Shen *et al* study. Whilst that study did indeed analyse strains with a SCRaMbLEd short circular chromosomal arm, they did not perform any phenotype-based selection, or improve or explore any particular phenotype at all. They were also not able to resolve all of the SCRaMbLEd circular DNAs with the Illumina-based techniques used. Thus, we are confident that our work is sufficiently original as it represents (i) the first time that SCRaMbLE has been demonstrated as a strain improvement technique for enhancing targeted phenotypes, (ii) the first time that SCRaMbLE has been combined with the expression of heterologous pathways and phenotypes, (iii) the first time that specific full chromosome-scale SCRaMbLE events have been identified and (iv) the first time that nanopore sequencing technology has been used to identify SCRaMbLE events. In the revised manuscript we

now adjust the introduction text to clarify how this study is designed to advance upon previous work (**Lines 50-67**).

As mentioned above, in revision, we have now fully genome-sequenced VB2 using nanopore reads and identified the SCRaMbLE events occurring in this strain (see **Figure 3**). Although here we have only used sequencing here to look at these two example strains, the purpose of using nanopore sequencing in this work was intended to (1) confirm that SCRaMbLE events have occurred in our improved strains and, (2) demonstrate that SCRaMbLE events in strains of interest can be effectively identified via nanopore sequencing using our described workflow. Our sequencing work was not done as a confirmation that SCRaMbLE can occur in general, as this was shown in previous work.

Major points

1. The high copy number of 2-micron plasmid is due to partitioning and amplification systems of the plasmid. An increasing number of host-encoded factors are found to be involved in the faithful segregation of the 2-micron plasmid to support these systems. Therefore, the authors should analyze the post-SCRaMbLEd chromosomes by deep sequencing and try to identify host factor mutations (or arrangement) present in the VB2 host involved in increased expression of genes placed on 2-micron plasmids.

We thank the reviewer for these comments. We have now expanded our characterisation of VB2 by fully sequencing the genome, revealing two SCRaMbLE events: deleting UBP3 and inverting the SWI4/LSM4 region. This new experimental work is shown in **Figure 3** in and described in the main results text in **Lines 230-235**. While not a full elucidation of the molecular mechanisms, this at least sheds more light on one of the examples generated by the SCRaMbLE approach our manuscript describes. In the discussion section we speculate on a possible mechanism that could increase 2-micron copy number through the chromosomes rearrangements seen in VB2 (**Lines 374-381**).

*2. Strain XD4 showing improved growth in xylose was isolated, but poorly characterized. The only analyses that the authors performed beside DNA sequencing was to measure growth rate and GCN4 transcript level (negative result). Inability of *S. cerevisiae* to grow in xylose could be due to glucose repression, slow xylose transport, cofactor imbalance in the xylose reductase/xylitol dehydrogenase pathway, repression of a heterologous xylose isomerase, the low efficiency of downstream pathways and low ethanol production (Hou et al. FEMS Yeast Res. 2017). The authors should address these possibilities in XD4.*

To provide additional further characterisation of XD4, we have now performed these suggested experiments and found that the slow growth of synV in xylose is due to an accumulation of xylitol, symptomatic of a cofactor imbalance affecting the xylitol dehydrogenase-mediated step of the heterologous xylose metabolism pathway. This accumulation is not evident in XD4, indicating that the redox cofactor imbalance has been ameliorated in the SCRaMbLEd strain. This new experimental work is provided in **Figure 2** and described in the main text (**Lines 183-191**).

3. The authors used synV, a haploid yeast strain in which the natural chromosome V has been replaced with a synthetic version containing the SCRaMbLE system, hoping gene arrangement in synV. But they never showed that the causative genetic perturbation is in fact on synV and not on other chromosomes (or not on the plasmid in the case of engineering xylose fermentation).

We agree with the reviewer that it is important to discount mutations on the pJCH006 plasmid as being responsible for the fast xylose growth phenotype of XD4. To do this, we cured XD4 of pJCH006 by serial growth on rich glucose media and then retransformed the plasmid back in from the original stock of pJCH006. The retransformed XD4 strain retained the fast xylose growth phenotype, demonstrating that the phenotypic enhancement is not due to plasmid mutation. This new work is described in **Lines 178-181**. Similar work was also done for VB2 as described in the main results text (**Lines 93-95**).

Regarding the other chromosomes, sequencing of both XD4 and VB2 do not reveal any evident changes (e.g. copy number variations, rearrangements) to the non-synthetic chromosomes of either strain. Therefore we are confident that the SCRaMbLE events on SynV are the main contributor to this phenotype, although in the revised manuscript discussion we now acknowledge that elucidating the mechanisms to explain the phenotype from the genetic changes is likely to be non-trivial and the topic of further studies (**Lines 370-373 & 389-391**).

4. Only single colony was picked up during the screening and further analyzed. To evaluate this new synthetic genome approach, it is necessary to analyze more colonies and to address the frequencies to generate improved strains.

We thank the reviewer for this comment. To address this point, we have now further conducted additional SCRaMbLE experiments to evaluate the frequency at which sfGFP expression from a 2-micron plasmid can be improved, and also how this is affected by the length of induction. Our results indicate that our experience of finding improved strains in every round of 96-well colony screening performed is consistent with the rates seen in these larger scale experiments. These new data are presented in **Figure 4** and described in a new results section (**Lines 266-314**).

Minor points

*1. In page 2, line 43, they mentioned “now provides a new *S. cerevisiae* host strain capable of doubling expression of pathways and genes placed on 2-micron plasmids”. But how about the plasmid harboring sfGFP? The increase of sfGFP is only ~1.5.*

The text has been amended to give the more accurate range of 1.6-2.3x.

2. Why was the fluorescence output from LEU2-2-micron plasmid lower than that from URA3- 2-micron plasmid?

Plasmids with a 2 micron replicon and a LEU2 marker have been shown in the literature to have a lower copy number than 2 micron plasmids with other auxotrophic markers, such as URA3 (Karim et al 2013 doi: 10.1111/1567-1364.12016). The text has been modified at **Lines 107-108** to explain this and cite the past work.

3. There is no description how to make pBAB plasmids (pBAB011, pBAB012, pBAB015, pBAB016).

We thank the reviewer for spotting this and apologise for the omission. The methods section has been updated with the missing information (**Lines 432-434**).

4. They measured the 2-micron plasmid copy number with pJCH017. How about the copy number of 2-micron plasmid harboring sfGFP? Based on the expression level of sfGFP, the copy number of pBAB011 could be ~1.5.

We have modified the text to change the claim from “double” increase/copy number to a more accurate 1.6x to 2.3x.

5. Regarding growth of XD4 in SCX URA-, why is there so long lag time?

A lag in engineered *S. cerevisiae* strain growth on xylose is commonly seen in the literature, including in a recent study by Verhoeven *et al* (doi:10.1038/srep46155). In the yeast *Yarrowia lipolytica*, long lag times are also seen during xylose growth (e.g. Li *et al*, doi: 10.1002/biot.201600210) and a recent study showed that these can be reduced by overexpression of the pathway enzymes (Ledesma-Amaro *et al*, <https://doi.org/10.1016/j.ymben.2016.07.001>). However, as the xylose utilisation pathway is itself not subject to SCRaMbLE in our experiments, these enzymes cannot be overexpressed, and so the presence of a lag time in growth is not surprising.

6. In page 4, line 19, did they use 100 ml culture or 60 ml culture?

Thank you, this was an error. The number has now been changed to the correct value of 60 ml.

7. There are two issues going on in the second round of SCRaMbLE. The first is loss of viability during induction of genome rearrangement. The second is failure to obtain improved strain after direct selection on SCX URA-agar plate. The author mixed up these two.

The reviewer correctly points out that both of these factors were at play during the second round of SCRaMbLE. We have now rewritten this to make this more clear (**Lines 310-314**).

8. In page 4, line 39, they mentioned that illumina-based sequencing was not able to resolve all sampled strains. It is true, but the previous study reported that simple genome arrangement can be easily analyzed by short-insert sequencing. Did the improved strain analyzed in this study have too complex rearrangements?

Our experience with sequencing indicates that the larger-scale events, such as the 7kb inversion, are difficult to identify with shorter read-length sequencing technologies. This is compounded by the fact that every SCRaMbLE event occurs at a 34 bp loxPsym site rather than at a chromosomal location with a unique sequence. Although it may be possible to identify smaller events such as deletions using short-insert sequencing, the ability to confidently identify these and much larger events (e.g. 150 kb inversions) in single nanopore reads means that nanopore technology generates data that allows higher degrees of certainty for recombination events. As a result of this, the Sc2.0 community is currently moving towards long read sequencing technology as standard for SCRaMbLE analysis and is moving towards nanopore sequencing as a preferred method.

9. The authors mentioned that the improved host strains had no obvious changes in cell morphology (Figure S2, Figure S10). But compared with other quantitative data, quality of cell morphology data is quite weak. Pictures with one or two cells are not evidential. It is necessary to analyze the difference statistically.

We entirely agree with the reviewer that the cell morphology data were not up to the standard of the other experimental data. We have now repeated the data collection and analysis to a higher standard in the revised work. These improved images and analysis are now in the Supplementary Materials (**Figures S2, S3, S11 and S12**).

10. Top yeast cell in Fig. 1a contains cre plasmid, but bottommost cell dose not. Did author introduce any plasmid curation step during screening?

We have found that SCRaMbLEd cells very readily lose the cre plasmid once the appropriate auxotrophic selection is removed. We have never found colonies picked from the initial SCRaMbLE plates that have retained the cre vector and so a separate plasmid curation step is not usually necessary. We have now mentioned this in the text (**Lines 85-86**).

11. In Fig. 1c including "Inset", is the culture normalized for OD600 or OD700?

The methodology of Fig 1c could indeed be clearer. In the inset image, culture was normalised for biomass using OD₇₀₀ and the figure legend erroneously stated OD₆₀₀. This has now been corrected.

Reviewer #4:

Remarks to the Author:

Blount et al. utilise the Synthetic Chromosome Recombination and Modification by LoxP-Mediated Evolution (SCRaMbLE) system towards a metabolic pathway evolution application to improve production host performance. Impressively, the authors showed that flux through two separate pathways could be improved through a simple experimental workflow using a yeast strain with a SCRaMbLE chromosome V; specifically they increased violacein biosynthesis and xylose utilisation. As impressive as these results are, there are several criticisms that prevent my support in immediate publication.

Most critically, it is unclear from the described results whether the net result of the SCRaMbLE experiments is equivalent to screening a knockout collection. While this methodology is capable of producing inversions or large scale duplication events, there is no evidence that these genotypes provide a phenotypic advantage over simple deletions.

We thank the reviewer for their constructive comments, questions and suggestions. While screening a knockout collection is indeed a powerful and established approach, SCRaMbLE has several advantages over it, whilst also providing a complimentary alternative. In terms of advantages, the

amount of time and work involved in the process of generating and screening a pathway in all single knockouts is much greater than putting a pathway plasmid into one synthetic chromosome strain and then triggering the SCRaMbLE system and screening colonies. For example, to identify our strain with improved xylose growth, we only had to transform a plasmid into one strain, induce for a few hours, plate and then screen around 80 colonies. To achieve the same result with a knockout collection, the plasmid would need to be transformed into each of the thousands of individual members of the collection and each of those strains would need to be screened. Furthermore, as shown in Shen et al. (doi:10.1101/gr.193433.115) SCRaMbLE also can readily generate multiple knockouts (pairs of genes or more) which is beyond most knockout collections, and also generates duplications. We have now revised the discussion text to include a new table (**Table 1**) which compares SCRaMbLE to existing alternative strain diversification techniques for yeast, including knock-out libraries.

The sequenced data of the violacein pathway hit was not described

To address this, we have now fully sequenced the VB2 strain and identified SCRaMbLE events deleting UBP3 and inverting the SWI4/LSM4 locus. These new data are presented in **Figure 3** and described in the results text (**Lines 219-221 & 230-235**).

...and the functional genotype modification of the xylose utilising strain was not experimentally verified. For the latter, it seems likely that the excision of MXR1 is responsible for the improved growth, but the authors did not do the experiment to generate this deletion in a clean background strain.

We have now further characterised the XD4 strain to show that it does not suffer from a xylitol accumulation that is evident in the parental strain (see **Figure 2**), indicating that a redox cofactor imbalance has been ameliorated in XD4. In the new discussion section (**Lines 382-388**) we now speculate how this could be caused the identified chromosome rearrangements in XD4. We note that the MXR1 deletion could only partially-recreate improved growth on xylose in our experiments (see **Lines 241-249**), so the phenotype is therefore more than just an equivalent to a knockout. However we do not extend the work to a full elucidation of the mechanisms involved as this would ideally require major further experimental work (e.g. proteomics and metabolomics) that we feel goes beyond the scope of the paper.

It is feasible that other effects from inversions could have a beneficial effect, but this should be demonstrated to distinguish this methodology from other existing methods like deletion collection or CRISPRi-based screening.

Our new additional sequencing of the VB2 strain (see **Figure 3**) also reveals an inversion and deletion. In this case we expect that the inversion is primarily responsible because the UBP3 deletion alone is not sufficient to replicate the VB2 phenotype (see **Lines 241-249**), and because of the known roles SWI4 has in DNA replication and cell cycle control (see **Lines 376-381**).

The SCRaMbLE method is capable of producing different types of recombination event within a cell, and this mixture of deletions, inversions, translocations and duplications leads to a far larger potential phenotypic space than that seen with methods such as deletion collections and CRISPRi. We do not envisage SCRaMbLE supplanting all existing methods but rather to be a useful strain engineering

tool, with particular strengths in the speed and ease of the process and the large genotypic space sampled. We have now revised our manuscript to better emphasise that SCRaMbLE provides a new tool for strain diversification and to better qualify the scope of our work, which is to demonstrate the methodology. We have also now added **Table 1** to compare SCRaMbLE-based diversification to other methods for strain diversification, including knock-out collections.

Similarly, additional experiments exploring the mechanism for improvement in these two pathways would improve the manuscript. The increase in violacein production as a result of increased expression from a 2 micron plasmid is possibly interesting and broadly impactful. Determining the mechanistic source of this improvement would potentially allow translation of this improvement to other yeast strains.

We apologise that the initial version of the manuscript did not clearly describe the intended scope of the manuscript. We have now revised the introduction text to clarify that the manuscript aim is to demonstrate SCRaMbLE as a new technique to rapidly generate isolates with improved phenotypes (**Lines 57-70**). While it would be interesting, we feel that fully determining and understanding the molecular mechanisms at play in the strains generated here is beyond the scope of this paper. Indeed, one would expect such work to be a study in and of itself, especially given that it is well-established that the links between genotype and phenotype for metabolic systems are often highly non-linear.

Having said that, in the revised work we have now made further efforts to better-characterise the strains, and new results have been added that offer more insights into the xylose growth phenotype (see **Figure 2**). We have also now sequenced the VB2 phenotype (see **Figure 3**), and in the discussion section we speculate how the identified rearrangements may explain the increased copy number of 2 micron plasmids in this strain (**Lines 376-388**).

Another difficulty in interpreting the generalizability of using SCRaMbLE for metabolic pathway improvement is the lack of statistics for how frequently hits are identified. In both experiments, a single hit was isolated in a sample size of approximately 90 strains. This success is surprising and impressive - however, how representative is this? Since it is only one hit, one can't estimate. For these two pathways, if the sampled set was 10-fold larger, would other hits be isolated? The ease of the SCRaMbLE workflow should enable this question to be answered. If additional hits are obtained, would they show the same functional genotype modification?

We agree that further characterisation of the SCRaMbLE process and the frequencies at which improved strains arise greatly improves the manuscript. However, it is difficult to exhaustively test the utility of SCRaMbLE as every bespoke problem will have a different set of solutions, in terms of SCRaMbLE events helping that phenotype. *i.e.* there may be many more ways that SCRaMbLE might improve Pathway X versus Pathway Y.

To try to address this point, we felt that SCRaMbLE-ing synV cells with sfGFP expression encoded on a 2-micron vector and screening resultant colonies for GFP output would be a good proxy for what one might expect. These new experiments indicated that screening at a 96-well level of throughput is sufficient to isolate improved strains with standard SCRaMbLE induction times. These new data are presented in **Figure 4** and described in a new results section (**Lines 266-314**).

Next, colonies that showed enhanced sfGFP expression from a 2 micron vector were isolated and screened for the SCRaMbLE events identified by VB2 sequencing in Figure 3. As none of the strains

were found to have a UBP3 deletion or a SWI4/LSM4 inversion, this was an indication that different SCRaMbLE events were responsible for the enhanced phenotypes. This is not surprising, as the potential genotypic space arising from the SCRaMbLE of a chromosome with 174 symmetrical recombination sites is huge and so in most cases we would expect different solutions to phenotype enhancement to arise.

Furthermore, some phenotypes, like xylose utilisation, are amenable to selection or high-throughput screening. Was there a reason to choose 87 colonies without selective pressure rather than performing a large scale selection by growing a much larger population on xylose media?

A great question. The reviewer is correct in pointing out that if we were to set out to achieve the highest xylose utilisation rate that we could, selective pressure of xylose growth would have been by far the easiest, and probably most effective, way of achieving this. However, as the purpose of this work was to describe the methodological approach and demonstrate the wider utility of the technique, we wanted to demonstrate that a 96-well screening process would be sufficient to isolate improved strains and a selective pressure (which would be unavailable for many heterologous pathways) is not necessary. In the revised manuscript we have now clarified our reasoning for this approach of screening rather than selective pressure, and for the colony numbers chosen (**Lines 266-271**). It is also worth noting that as no selective pressure was placed on the cells prior to screening, we intentionally minimised the risk of low-frequency off-target mutations occurring that may have affected the phenotype (i.e. non-SCRaMbLE events).

REVIEWERS' COMMENTS:

Reviewer #2 (Remarks to the Author):

The authors responded to almost all suggestions and comments, prepared new Figures and Tables, extensively improved the manuscript, and therefore, I am satisfied with the revision.

Reviewer #3 (Remarks to the Author):

I am satisfied with the authors' revision and responses to my comments.

Reviewer #4 (Remarks to the Author):

I think the manuscript has been considerably improved and is now ready for publication. I also thank the authors for their clear and attentive responses to the reviewers' questions/comments.